# CASCADE Your Datasets for Cross-Mode Knowledge Retrieval of Language Models

**Runlong Zhou**[*]
University of Washington
vectorzh@cs.washington.edu

**Yi Zhang**
xAI
zhayi0928@gmail.com

## Abstract

Language models often struggle with cross-mode knowledge retrieval – the ability to access knowledge learned in one format (mode) when queried in another. We demonstrate that models trained on multiple data sources (e.g., Wikipedia and TinyStories) exhibit significantly reduced accuracy when retrieving knowledge in a format different from its original training mode. This paper quantitatively investigates this phenomenon through a controlled study of random token sequence memorization across different modes. We first explore dataset rewriting as a solution, revealing that effective cross-mode retrieval requires prohibitively extensive rewriting efforts that follow a sigmoid-like relationship. As an alternative, we propose CASCADE, a novel pretraining algorithm that uses cascading datasets with varying sequence lengths and computing losses on only the second half of each training sequence to capture knowledge at different scales. Our experiments demonstrate that CASCADE outperforms dataset rewriting approaches, even when compressed into a single model with a unified loss function. This work provides both qualitative evidence of cross-mode retrieval limitations and a practical solution to enhance language models' ability to access knowledge independently of its presentational format. To facilitate research in the field of LLMs, the code is publicly released.[1]

## 1 Introduction

Large language models (LLMs) are often pretrained on corpus comprised of several sources, each with a unique *mode* (wording style, organization format, etc., will also be referred to as *format*). Although LLMs can achieve low losses on validation sets from the same mode, we observe concrete examples that they cannot perform *cross-mode knowledge retrieval* effectively. For example, we can pretrain a language model on both Wikipedia excerpts and the TinyStories (Eldan & Li, 2023) dataset until convergence. However, when we query the model for knowledge present in the Wikipedia training set *using a story format*, the generated response shows surprisingly low accuracy on average. We illustrate this in Figure 1, with details in Appendix B.

Motivated by this phenomenon, we research the following question:

> *How can we make language models capable of cross-mode knowledge retrieval?*

We approach this problem quantitatively, focusing on a toy yet fundamental task of memorizing *random token sequences* in different modes (Wikipedia and TinyStories). Memorization of random token sequences can be precisely quantified by computing log probabilities. We investigate whether language models learn *spurious correlations* between **knowledge** and **mode** instead of learning knowledge independently. To the best of our knowledge, while spurious correlations in natural language processing have been widely studied in classification tasks, **they remain underexplored in general language modeling tasks, particularly**

---

[*]Part of this work done when Runlong was an intern at Microsoft Research, Redmond.
[1]https://github.com/zhourunlong/CASCADE_public

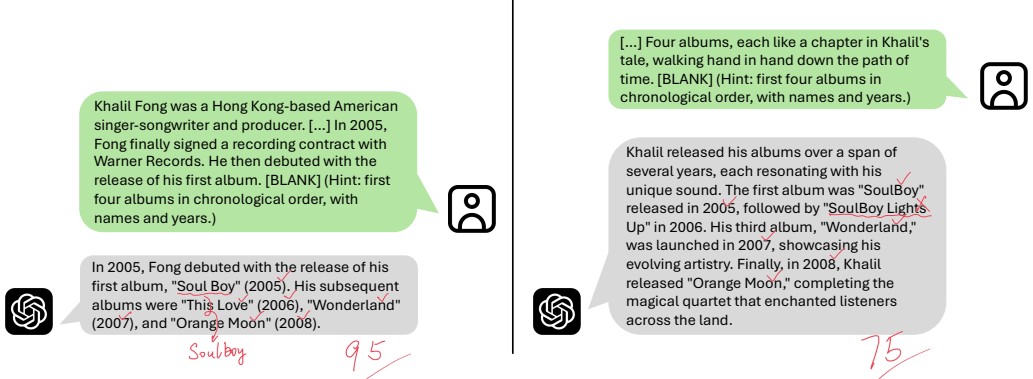

Figure 1: GPT-4o shows inconsistent accuracies when prompted with the same question but in different formats. Left: the query is in a Wikipedia format. Right: the query is in a story format. Find more detailed examples in Tables 3 to 5.

*in knowledge memorization and manipulation*. We hope this work will serve as an initial study of spurious correlations in general language modeling and inspire more effective and efficient methods to alleviate this issue.

## 1.1 Our contributions

Our contributions are twofold - both qualitative and quantitative.

**Qualitatively**, we build a pipeline (Appendix B) that demonstrates how LLMs fail at *cross-mode knowledge retrieval*.

**Quantitatively**, we focus on the *pretraining* stage to improve language models' ***cross-mode knowledge retrieval*** capability. Our quantitative contributions can be summarized as follows:

• **Dataset rewriting.** We first study how the *ratio* between non-cross-mode (original in the dataset) and cross-mode data (*rewritten* into the dataset, with occurrences controlled by us) affects the evaluation performance of cross-mode knowledge retrieval (Section 4). We plot curves of evaluation performance with respect to the ratio $r$ between same-mode and cross-mode knowledge occurrences. These curves follow a *sigmoid*-like function: $f(r) = a \cdot \sigma(b(\log(r) - c))$. These results demonstrate that ***effective cross-mode knowledge retrieval requires extensive rewriting effort, which is prohibitive in practice***.

• **Novel algorithm: CASCADE.** We propose a novel algorithm, CASCADE, as a solution. During pretraining, we use a series of ***cascading*** datasets with *different sequence lengths* to help the language model *capture knowledge at different scales*. The loss is calculated on only the second half of each training sequence. We first show that an original form of CASCADE using model ensemble achieves better performance than dataset rewriting (Section 5.1), then improve its complexity by compressing it into a single model with a single loss function (Section 5.2). We also visualize how different sequence lengths contribute to completing different knowledge.

## 2 Related works

We discuss the most related line of works here, deferring the other works to Appendix A.

**Consistency.** Consistency in language models has earned significant research attention. Elazar et al. (2021) defined consistency as "invariance under meaning-preserving alternations" and introduced PARAREL for evaluating factual knowledge consistency across paraphrases. Inconsistency manifests across various NLP applications: Ribeiro et al. (2019) identified inconsistencies in question answering systems, while Kryscinski et al. (2019) studied factual consistency in summarization. Li et al. (2019) and Camburu et al. (2019)

examined inconsistencies in natural language inference (NLI) systems and explanations, respectively. Researchers have proposed various improvement approaches: Elazar et al. (2021) introduced a consistency loss function, Kassner et al. (2021) proposed augmenting PLMs with an evolving memory, Chen et al. (2021) developed explanation-based post-training, and Asai & Hajishirzi (2020) utilized data augmentation with symmetricity properties.

*We highlight that while at a high level the issues associated with **cross-mode knowledge retrieval** could be classified as inconsistency, they differ drastically. In previous consistency studies, input changes are typically small perturbations such as synonym replacement, word or sentence permutation, or statement-QA conversion, leaving word styles largely unchanged. In contrast, **cross-mode knowledge retrieval** applies to entirely different text sources with highly diverse word styles, making the language model more prone to derive spurious correlations between the mode and the knowledge.*

## 3 Settings

In this work, we study the knowledge memorization mechanism in language models. Specifically, we care about ***how much will the format text influence the language model's memorization of the knowledge***, and ***how to reduce this influence***. To this end, we will construct datasets that admit a well-defined criterion of the extent of memorization. The high-level idea is to define knowledge pieces as *random token sequences*, thus any language model is said to memorize the knowledge only if it can perfectly *generate the whole sequences*, admitting log probabilities as quantification of memorization. The *modes* or *formats* are defined as texts from different datasets. The language models should separate knowledge from modes to perform well on cross-mode knowledge retrieval tasks.

**Tokenization.** We process everything in the *token space*. We use the GPT-2 tokenizer (`tiktoken.get_encoding("gpt2")`) in this study, which has a token range of $[0, 50256]$. Some other tokens may be used in the experiments, and we constrain them to be in a *separate* range of $[50257, 50303]$.

**Notations.** Denote $\Sigma$ as the set of all possible tokens. We use subscripts to denote the mode name, superscripts to denote the index in a set, and numbers in brackets to denote the index in a set. For a set $\mathcal{X}$, we use $|\mathcal{X}|$ to denote the number of unique elements in $\mathcal{X}$. For a sequence $a$, we use $|a|$ to denote the length of $a$.

**Indexing.** We follow Python's indexing convention. Numerical indices start from 0. When using a range to index, the lower bound is included while the upper bound is *excluded*. When indexing an array $a$, a lower bound of 0 or an upper bound of $|a|$ can be omitted. A *negative* index $a[-i]$ means $a[|a| - i]$.

### 3.1 Core concepts

First, we introduce core concepts that will be referenced frequently when constructing datasets and during training and evaluation.

**Formats/Modes.** We use existing datasets, English Wikipedia excerpts and TinyStories (Eldan & Li, 2023), as *format/mode texts*. They are denoted as $\mathcal{F}_{\mathsf{wiki}}$ and $\mathcal{F}_{\mathsf{ts}}$, respectively. For training, we take portions from each format, making them roughly equal in token counts. We take disjoint portions from each format for evaluation.

**Knowledge.** We use ***random token sequences*** as *knowledge* for the following reasons:

• **Quantification:** Memorizing random token sequences requires precise token-by-token memorization, unlike general knowledge which can be rephrased in various ways. This enables exact quantification by computing the log probability of generating a desired random token sequence.

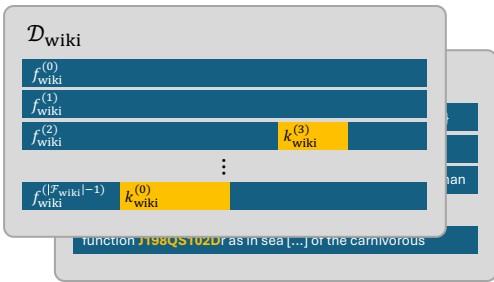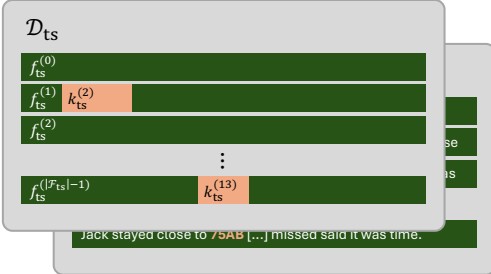

Figure 2: Illustration of the datasets. Each line represents a block of $L_{\text{blk}}$ consecutive tokens in $\mathcal{F}_{\text{wiki}}$ or $\mathcal{F}_{\text{ts}}$. Knowledge pieces overwrite to some of the blocks in arbitrary positions. The de-tokenized datasets are shown in the background.

• **Exclusiveness:** We can ensure these knowledge pieces neither appear in mode texts nor correlate with each other. This prevents knowledge leakage in the training set and eliminates correlation between mode and knowledge.

We construct $K = 32$ pieces of knowledge for each mode:

$$\mathcal{K}_{\text{wiki}} = \{k_{\text{wiki}}^{(0)}, k_{\text{wiki}}^{(1)}, \ldots, k_{\text{wiki}}^{(K-1)}\}, \quad \text{and} \quad \mathcal{K}_{\text{ts}} = \{k_{\text{ts}}^{(0)}, k_{\text{ts}}^{(1)}, \ldots, k_{\text{ts}}^{(K-1)}\}.$$

Each piece of knowledge $k \in \mathcal{K}_{\text{wiki}} \cup \mathcal{K}_{\text{ts}}$ is a random token sequence with length between $\underline{L_{\text{knw}}} = 8$ and $\overline{L_{\text{knw}}} = 512$ (both inclusive), and the tokens are from the range of $[50296, 50303]$. Each position in the sequence is independently sampled from a uniform distribution over the token range. To make knowledge exclusive to its corresponding format, these two knowledge sets are *disjoint at the sequence level*: $\mathcal{K}_{\text{wiki}} \cap \mathcal{K}_{\text{ts}} = \varnothing$.

**Queries.** Queries are "hints" for the language model to complete a knowledge piece, so we set them as *prefixes* of each knowledge piece. To make the problem well-defined, the prefixes should be unique so that they correspond to knowledge pieces in a one-to-one manner. We find the shortest prefix length so that the induced queries are different:

$$\ell = \min l \quad \text{such that} \quad |\{k[0:l] \mid k \in \mathcal{K}_{\text{wiki}} \cup \mathcal{K}_{\text{ts}}\}| = 2K.$$

The queries are defined as

$$\mathcal{Q}_{\text{wiki}} = \{q_{\text{wiki}}^{(i)} := k_{\text{wiki}}^{(i)}[0:\ell] \mid 0 \leqslant i < K\}, \quad \text{and} \quad \mathcal{Q}_{\text{ts}} = \{q_{\text{ts}}^{(i)} := k_{\text{ts}}^{(i)}[0:\ell] \mid 0 \leqslant i < K\}.$$

### 3.2 Problem formulation

Now we formally describe our problem of interest: ***cross-mode knowledge retrieval***.

**Datasets.** There are two fixed datasets, $\mathcal{D}_{\text{wiki}}$ and $\mathcal{D}_{\text{ts}}$, each containing knowledge from only one mode (itself). Taking Wikipedia as an example: The format texts from $\mathcal{F}_{\text{wiki}}$ are divided into consecutive blocks with length $L_{\text{blk}} = 1024$. We set hyperparameter $N_{\text{occ}} = 8192$ as the number of occurrences[2] for each knowledge piece $k \in \mathcal{K}_{\text{wiki}}$ in $\mathcal{F}_{\text{wiki}}$, totaling $KN_{\text{occ}}$ occurrences of knowledge pieces. These knowledge pieces are distributed across $KN_{\text{occ}}$ blocks sampled uniformly. Inside each block $f_{\text{wiki}}$, the knowledge piece overwrites a random, consecutive subsequence with equal probability. An illustration is shown in Figure 2.

**Evaluation.** Given these two datasets, we want to quantify the ***cross-mode knowledge retrieval*** capability of language models. Since the only way to memorize the random sequences is to perfectly generate them, the task is to do *completion* on the remaining tokens given a query ("hint") $q$. We set $N_{\text{occ}}^{\text{test}} = 16$ occurrences for each query $q \in \mathcal{Q}_{\text{wiki}} \cup \mathcal{Q}_{\text{ts}}$. For each $q$, we randomly sample $N_{\text{occ}}^{\text{test}}$ blocks of length $L_{\text{blk}}$ from the evaluation portion of *each* format, $\mathcal{F}_{\text{wiki}}$ and $\mathcal{F}_{\text{ts}}$, and overwrite it to the *end* of each block. Suppose $q = k[0:\ell]$ for

---

[2]This satisfies the 1000-exposure requirement in Allen-Zhu & Li (2024d).

some $k \in \mathcal{K}_{\mathsf{wiki}} \cup \mathcal{K}_{\mathsf{ts}}$ by our construction, $|k| = L_{\mathsf{knw}}$, and the format text block is $f$ such that $|f| = L_{\mathsf{blk}}$. The criteria is the normalized log probability of the completion part:

$$\frac{1}{L_{\mathsf{knw}} - \ell} \sum_{i=\ell}^{L_{\mathsf{knw}}-1} \log \mathcal{M}_\theta(k[i] \mid f[:-L_{\mathsf{knw}}], k[:i]),$$

where $\mathcal{M}_\theta$ is the model parameterized by $\theta$. An illustration can be found in Figure 3.

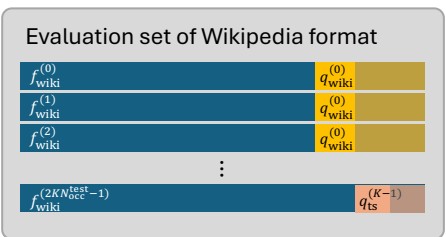
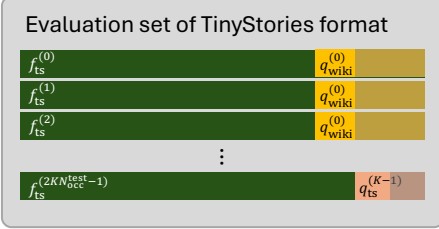

Figure 3: Illustration of the evaluation datasets. The shadowed parts are for completion and log probability calculation. The format texts are taken from the *evaluation* split of $\mathcal{F}_{\mathsf{wiki}}$ and $\mathcal{F}_{\mathsf{ts}}$, so they are different from those in Figure 2.

## 4 A straightforward approach: rewrite the datasets

Direct training on $\mathcal{D}_{\mathsf{wiki}} \cup \mathcal{D}_{\mathsf{ts}}$ yields poor performance – as shown in Figure 5, where dashed horizontal lines represent normalized log probabilities of completions after direct training using $\mathcal{D}_{\mathsf{wiki}} \cup \mathcal{D}_{\mathsf{ts}}$. Qualitative results (Appendix B) also support this argument. The language model likely learned a spurious correlation between mode and knowledge, so when queried with $f_{\mathsf{wiki}} \, q_{\mathsf{ts}}$ or $f_{\mathsf{ts}} \, q_{\mathsf{wiki}}$, it fails to correctly complete with $k_{\mathsf{ts}}$ or $k_{\mathsf{wiki}}$.

### 4.1 Method description

A straightforward approach to reduce this spurious correlation is to rewrite the datasets, incorporating *cross-mode* knowledge. For example, when rewriting $\mathcal{D}_{\mathsf{wiki}}$ into $\mathcal{D}'_{\mathsf{wiki}}$, besides the original $KN_{\mathsf{occ}}$ occurrences of knowledge pieces in $\mathcal{K}_{\mathsf{wiki}}$, we set a hyperparameter $N^{\times}_{\mathsf{occ}}$ as the number of occurrences for each cross-mode knowledge in this dataset. In practice, identifying and rewriting all exclusive knowledge is costly, so we use only $k_{\mathsf{ts}}^{(0)}, \ldots, k_{\mathsf{ts}}^{(K/2-1)}$ to rewrite the dataset and use $k_{\mathsf{ts}}^{(K/2)}, \ldots, k_{\mathsf{ts}}^{(K-1)}$ as **hold-out** knowledge for evaluation. Each $k_{\mathsf{ts}} \in k_{\mathsf{ts}}^{(0)}, \ldots, k_{\mathsf{ts}}^{(K/2-1)}$ appears exactly $N^{\times}_{\mathsf{occ}}$ times in $\mathcal{D}'_{\mathsf{wiki}}$, using the same method to generate $\mathcal{D}_{\mathsf{wiki}}$. An illustration can be found in Figure 4.

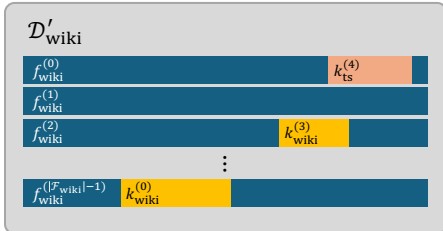
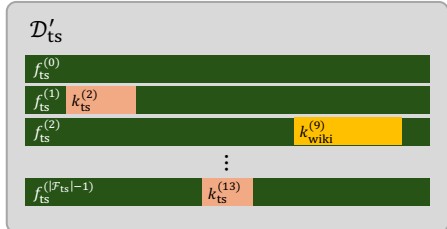

Figure 4: Illustration of dataset rewriting. Readers can compare with Figure 2.

For notational ease, we use the following shorthand:

• $f_{\mathsf{ts}} \, q_{\mathsf{ts}}$ and $f_{\mathsf{wiki}} \, q_{\mathsf{wiki}}$: evaluation data with a query from the same mode as the format text.

• $f_{\mathsf{ts}} \, q_{\mathsf{wiki}}$ and $f_{\mathsf{wiki}} \, q_{\mathsf{ts}}$: evaluation data with a cross-mode query.

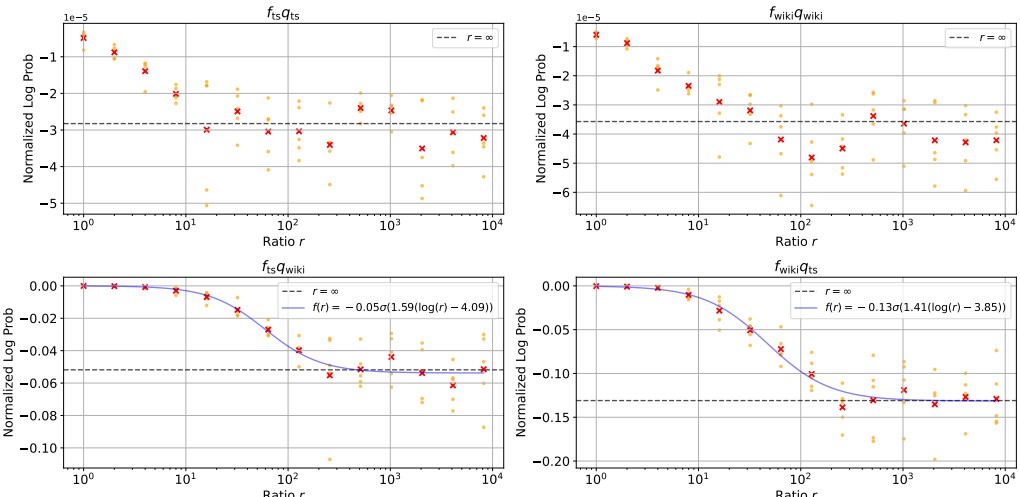

Figure 5: Normalized log probabilities for different ratios. The $x$-axis is in log scale. Yellow dots represent results from individual runs with 5 random seeds, while red crosses show average values. Dashed horizontal lines indicate results from direct training on the original datasets, $\mathcal{D}_{\mathsf{wiki}}$ and $\mathcal{D}_{\mathsf{ts}}$. Cross-mode evaluations use only hold-out queries.

For example, in Figure 3, the first and last entries in the left part are denoted as $f_{\mathsf{wiki}}\, q_{\mathsf{wiki}}$ and $f_{\mathsf{wiki}}\, q_{\mathsf{ts}}$, respectively.

## 4.2 Results

We test this method's effectiveness by sweeping over $N_{\mathsf{occ}}^{\times} \in \{0\} \cup \{2^i \mid 1 \leqslant i \leqslant 13\}$. With ratio $r = N_{\mathsf{occ}}/N_{\mathsf{occ}}^{\times}$, we plot the relationship (Figure 5) between $r$ and the convergent values of normalized log probabilities in evaluation. Experiment details are deferred to Appendix D. A special case is $r = \infty$ (dashed horizontal lines), corresponding to $N_{\mathsf{occ}}^{\times} = 0$, which represents the scenario without rewriting.

We also report in Table 1 the normalized log probabilities for small ratios $r \in \{1.0, 2.0, 4.0\}$.

|  | $f_{\mathsf{ts}}\, q_{\mathsf{ts}}$ | $f_{\mathsf{wiki}}\, q_{\mathsf{wiki}}$ | $f_{\mathsf{ts}}\, q_{\mathsf{wiki}}$ | $f_{\mathsf{wiki}}\, q_{\mathsf{ts}}$ |
|---|---|---|---|---|
| $r = 1.0$ | $-4.87 \times 10^{-6}$ | $-5.94 \times 10^{-6}$ | $-6.75 \times 10^{-5}$ | $-2.98 \times 10^{-4}$ |
| $r = 2.0$ | $-8.78 \times 10^{-6}$ | $-8.80 \times 10^{-6}$ | $-1.93 \times 10^{-4}$ | $-9.25 \times 10^{-4}$ |
| $r = 4.0$ | $-1.39 \times 10^{-5}$ | $-1.82 \times 10^{-5}$ | $-7.76 \times 10^{-4}$ | $-2.21 \times 10^{-3}$ |

Table 1: Normalized log probabilities for small ratios, averaged over 5 random seeds.

## 4.3 Remarks

We now make several remarks on the method of dataset rewriting.

• **The relation between the log ratio and the cross-mode evaluation results roughly follows a sigmoid function.** We observe that a sigmoid-like function fits the points well, so we perform regression using:

$$f(r) = a \cdot \sigma(b(\log(r) - c)), \quad \text{where} \quad \sigma(x) = \frac{1}{1 + e^{-x}}.$$

The blue curves in Figure 5 display the regressed functions.

• **Meaningful results only come with extensive rewriting.** Table 1 shows that to achieve cross-mode query performance comparable to non-cross-mode queries, the ratio should be at most 4.0, meaning $N_{\mathsf{occ}}^{\times} \geqslant 2048$. However, even when $r = 1.0$ ($N_{\mathsf{occ}}^{\times} = 8192$), normalized log

probabilities for cross-mode queries remain at order $10^{-4}$, still one order of magnitude worse than non-cross-mode queries. Additionally, we rewrote half of the different knowledge pieces, resulting in rewritten knowledge of the same order as the original knowledge. In practice, such extensive rewriting requires significant human effort to identify and rewrite knowledge differently across contexts. Even with the help of language models, rewriting prompts need case-by-case design, which also demands substantial human effort. Moreover, we *hypothesize* that:

> ***Data with $n \gg 2$ independent modes requires rewriting all $n(n-1)$ pairs.***

If this hypothesis is true, dataset rewriting becomes prohibitively impractical for achieving satisfactory performance. Results in Table 11 support this hypothesis when $n = 3$.

## 5    A cure: CASCADE the datasets

As a starting point, we consider an easier problem: suppose the knowledge can only appear in the *end* of each sequence blocks of length $L_{\mathsf{blk}}$, and they all have the *same* lengths of $L_{\mathsf{knw}}$. Assume that $L_{\mathsf{blk}}$ is a multiple of $L_{\mathsf{knw}}$. If we want the model to perfectly memorize the knowledge without being affected by modes, we can use a context length of $L_{\mathsf{ctx}} = L_{\mathsf{knw}}$ in training. This guarantees that each piece of knowledge fits exclusively in some training sequence, so that *it is not correlated with any mode*.

In the problem described in Section 3.2, we know neither the exact position nor the exact length of knowledge pieces, making it impossible to fit them exclusively within training sequences. As an alternative design, we aim to ensure each knowledge piece occupies *a large portion* of some training sequence to minimize the influence of modes.

### 5.1    Capturing knowledge with doubling context lengths

Roughly speaking, for a knowledge piece of length $L_{\mathsf{knw}}$, if we set the context length $L_{\mathsf{ctx}} \leqslant 2L_{\mathsf{knw}}$, then regardless of its location in $\mathcal{D}$, it will occupy *at least half* of the tokens in some training sequence. This can be guaranteed when training sequences overlap by $L_{\mathsf{ctx}}/2$. Since we assume $L_{\mathsf{knw}} \leqslant \overline{L_{\mathsf{knw}}} = 512$, we can train a small number of language models with context lengths $8, \ldots, 1024 = L_{\mathsf{ctx}}$ using a series of ***cascading*** datasets (Figure 6, with details explained in Section 5.1.1). This ensures each knowledge piece is captured by *at least one* language model.

During evaluation and generation, we predict the next token using a probability distribution that is a weighted average over all models (after normalization).

Since one pass of length $L$ in a transformer requires $\Theta(L^2)$ time, and our context lengths follow a geometric sequence, using all models adds little computational overhead compared to using a single model. We elaborate on this idea in Appendix C.2.

#### 5.1.1    Training

Now we present a novel algorithm, ***Original CASCADE***, realizing the above high-level idea. "Original" here is to distinguish it from the compressed, more practical variant that will be introduced in Section 5.2.

We abuse the notation that the original dataset $\mathcal{D}$ is an array of tokens. Let $M = \log_2(2\overline{L_{\mathsf{knw}}}) = 10$. We train $M - 2$ models $\mathcal{M}_3, \mathcal{M}_4, \ldots, \mathcal{M}_M$. For each $3 \leqslant m \leqslant M$, $\mathcal{M}_m$ is trained on the dataset $\mathcal{D}_m$ with context length $L_{\mathsf{ctx}}^{(m)} = 2^m$, where $\mathcal{D}_m := \{\mathcal{D}[i \cdot 2^{m-1} : i \cdot 2^{m-1} + 2^m] \mid i = 0, 1, \ldots\}$. Note there are overlaps in the sequences of length $2^{m-1} = L_{\mathsf{ctx}}^m/2$.

For any training sequence $s \in \mathcal{D}_m$ with $|s| = 2^m$, the loss is computed ***only on the second half*** of the sequence, i.e., treating the first half as hint and the second half as completion:

$$\mathcal{L}_m(\theta) = \mathbb{E}_{s \sim \mathcal{D}_m} \left[ \sum_{i=2^{m-1}}^{2^m-1} - \log \mathcal{M}_\theta(s[i] \mid s[:i]) \right].$$

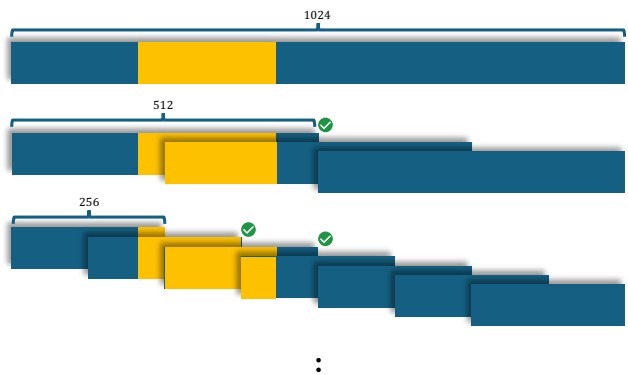

Figure 6: Illustration of *cascading* datasets. The highlighted part represents the knowledge. Blocks with a check mark in the top right indicate that the corresponding sequence captures the knowledge, satisfying Equation (1).

The intuition behind this choice is that we want language models to "think more" before they "speak." With access to the full context, models can predict future tokens more accurately. We show ablation results comparing ① non-overlapping sequences with full loss versus ② overlapping sequences with loss computed only on the second half in Table 2.

In practice, we use different batch sizes when training different models. Compared to direct training, we set $B_m = 2B \cdot L_{\text{ctx}}/L_{\text{ctx}}^{(m)}$, where the coefficient 2 accounts for the overlapping sequences. This batch size selection ensures that all models are updated for the same number of steps.

We show that each knowledge occurrence is guaranteed to be captured by some training sequence in Appendix C.1.

### 5.1.2 Model ensemble

Having trained $M - 2$ models $\mathcal{M}_3, \ldots, \mathcal{M}_M$, our next task is to ensemble them to produce valid probability distributions over tokens. Given an input token array $s$, we predict the next token by first querying each model with its corresponding context window to obtain $M - 2$ probability distributions: for $3 \leqslant m \leqslant M$, $x \in \Sigma$, $p_m(x) := \mathcal{M}_m(x \mid s[-2^{m-1} :])$.

We then define the confidence of each model by its maximum log probability across the token space: for $3 \leqslant m \leqslant M$, $c_m := \max_{x \in \Sigma} \log p_m(x)$. The weight of each model is calculated by: for $3 \leqslant m \leqslant M$,

$$w_m := \mathsf{Softmax}(m \mid \{-\log(-c_{m'})\}_{m'=3}^M) = \frac{\exp(-\log(-c_m))}{\sum_{m'=3}^M \exp(-\log(-c_{m'}))} \propto \frac{1}{-c_m}.$$

Considering the cases where $c_m$ is extremely close to 0, the practical implementation is $w_m \propto 1/(\epsilon - c_m)$ where $\epsilon = 10^{-9}$. The intuition is that we want to emphasize the predictions of models with high certainty while minimizing the influence of less confident models.

Finally, we compute the ensemble model using the weighted mixture of log probabilities as $l(x) := \sum_{m=3}^M w_m \log p_m(x)$, for any $x \in \Sigma$.

**Evaluation.** For efficient evaluation, we calculate probabilities for multiple tokens simultaneously with each model. Specifically, for $3 \leqslant m \leqslant M$, at position $i$, we input the sequence $s[i - 2^{m-1} : i + 2^{m-1}]$ to model $\mathcal{M}_m$ to compute logits for positions $i + 1, i + 2, \ldots, i + 2^{m-1}$, then increment $i$ by $2^{m-1}$. After obtaining logits from each model for all positions, we apply the ensemble method described above to calculate the final probability distribution.

### 5.2 Compressing all the models

While results in Table 2 demonstrate the effectiveness of using a series of *cascading* datasets, the increased total model size raises a significant concern. To address this issue, we compress

the models by training a single model $\mathcal{M}_\theta$ using the average of losses $\{\mathcal{L}_m\}_{m=3}^M$. We name this approach as *CASCADE*, which minimizes the *CASCADE* loss defined as:

$$\mathcal{L}_{\text{CASCADE}}(\theta) = \frac{1}{M-2} \sum_{m=3}^M \mathbb{E}_{s \sim \mathcal{D}_m} \left[ \sum_{i=2^{m-1}}^{2^m-1} -\log \mathcal{M}_\theta(s[i] \mid s[:i]) \right].$$

During evaluation or inference as described in Section 5.1.2, we replace all models $\mathcal{M}_m$ with the single model $\mathcal{M}_\theta$. This approach maintains the same model size as the baselines rather than being 8 times larger. Theoretically, CASCADE also does not incur higher time complexity as we explained in Appendix C.2.

## 5.3 Results

**To ensure fair comparison with dataset rewriting, we evaluate only using the *hold-out* knowledge for $f_{\text{ts}} q_{\text{wiki}}$ and $f_{\text{wiki}} q_{\text{ts}}$.** For a comprehensive analysis, we implemented both training configurations: non-overlapping training sequences with loss computed on the full sequence, and overlapping training sequences with loss computed only on the second half of each sequence.

**Ablation on practical running time.** In practice, the running time of a forward pass can be reduced significantly to an almost linear dependence on sequence length using FlashAttention (Dao et al., 2022; Dao, 2023). When training for the same number of epochs, the CASCADE loss requires approximately $M-2$ times the training time of the baseline method (direct training). For a fair comparison, we conducted an ablation study allowing the baseline method (with context length 1024) to train for the same duration as CASCADE.

We present the normalized log probabilities of (original) *CASCADE* and ablation studies in Table 2.

| Methods | | $f_{\text{ts}} q_{\text{ts}}$ | $f_{\text{wiki}} q_{\text{wiki}}$ | $f_{\text{ts}} q_{\text{wiki}}$ | $f_{\text{wiki}} q_{\text{ts}}$ |
|---|---|---|---|---|---|
| Direct Training (Ablation) | Non-overlap | $-1.93 \times 10^{-8}$ | $-1.43 \times 10^{-8}$ | $-4.77 \times 10^{-3}$ | $-1.53 \times 10^{-2}$ |
| | Overlap | $-2.29 \times 10^{-8}$ | $-2.16 \times 10^{-7}$ | $-2.66 \times 10^{-1}$ | $-4.31 \times 10^{-1}$ |
| Original CASCADE | Non-overlap | $-5.91 \times 10^{-6}$ | $-6.21 \times 10^{-6}$ | $-2.45 \times 10^{-5}$ | $-1.36 \times 10^{-4}$ |
| | Overlap | $-9.65 \times 10^{-9}$ | $-8.51 \times 10^{-9}$ | $\mathbf{\color{red}{-2.59 \times 10^{-8}}}$ | $\mathbf{\color{red}{-9.22 \times 10^{-7}}}$ |
| CASCADE | Non-overlap | $-3.87 \times 10^{-5}$ | $-3.95 \times 10^{-5}$ | $-1.87 \times 10^{-4}$ | $-1.54 \times 10^{-4}$ |
| | Overlap | $-3.26 \times 10^{-7}$ | $-3.44 \times 10^{-7}$ | $\mathbf{\color{red}{-3.71 \times 10^{-6}}}$ | $\mathbf{\color{red}{-5.06 \times 10^{-6}}}$ |

Table 2: Normalized log probabilities for different methods, averaged over 5 random seeds. Cross-mode results better than or equal to the order of $10^{-6}$ are in bold red text.

More results can be found in Appendix D.4: To better illustrate the contribution of different context lengths during evaluation, we display the normalized log probabilities when using only a single context length (*without model ensemble*) in Table 9, with specific context lengths *excluded* in Table 10, and visualize the weight vector $\{w_m\}_{3 \leqslant m \leqslant M}$ at each token position for various knowledge lengths in Figure 7. To show the generalization capability of CASCADE, we compare it with direct training and dataset rewriting when there are 3 modes (Table 11). We also show the performance of baselines when using a roughly double-sized model in Table 12.

## 5.4 Remarks

We now make several remarks for CASCADE.

• *Cascading* **the dataset substantially enhances the cross-mode knowledge retrieval capability of language models.** Results in Table 2 demonstrate that with *cascading* datasets, models achieve significantly improved cross-mode knowledge retrieval with overlapping sequences compared to non-overlapping sequences, and most importantly, outperform all baselines presented in Section 4.2. As anticipated, model compression introduces a minor performance degradation.

- **$\mathcal{L}_{\textbf{CASCADE}}$ functions as an implicit regularizer.** Unexpectedly, Table 9 shows that training with $\mathcal{L}_{\text{CASCADE}}$ alone, even without model ensemble, improves cross-mode knowledge retrieval capability. This loss function appears to implicitly regularize the language model against spurious correlations. Comparing Table 9 and the rows corresponding to CASCADE in Table 2, we observe that model ensemble further enhances performance by an order of magnitude.

- **Small context lengths are critical for initial positions.** Figure 7 illustrates that for the first few tokens in the completion, models with smaller context lengths exhibit greater prediction certainty. Additionally, the context lengths of 128 and 256 appear to be excessive according to Table 10.

- **CASCADE delivers benefits beyond simply increasing training epochs.** Rows corresponding to direct training in Table 2 confirm that merely extending training time does not enable baselines to match CASCADE's performance, with results on $f_{\text{ts}}\ q_{\text{wiki}}$ and $f_{\text{wiki}}\ q_{\text{ts}}$ remaining significantly inferior to CASCADE. Furthermore, in this ablation study, non-overlapping sequences notably outperform overlapping sequences. This occurs because cascading context lengths are essential for capturing "local" information, whereas using a single large context length and calculating loss only on the second half disrupts these "local" connections.

- **CASCADE generalizes to more modes better than dataset rewriting.** Table 11 confirms CASCADE's advantage over direct training and full rewriting when there are 3 modes. The row corresponding to full rewriting without one mode-knowledge pair corroborates the remark in Section 4.3 that omitting a mode-knowledge pair directly harms its evaluation performance, demonstrating that meaningful rewriting requires coverage of all pairs.

- **CASCADE's performance matches that of full rewriting while using less than half the model size.** Comparing Tables 11 and 12, increasing model size from 162M to 350M improves each method's performance by approximately one order of magnitude. With this increase, full rewriting (350M) matches CASCADE (162M), while direct training still struggles with cross-mode knowledge retrieval.

## 6   Conclusion

We investigated language models' cross-mode knowledge retrieval capability from both qualitative and quantitative perspectives. Our qualitative pipeline reveals that LLMs such as GPT-4o cannot perform cross-mode knowledge retrieval satisfactorily. Quantitatively, we formulated this problem using two format datasets as modes and random token sequences as knowledge, and experimented with a straightforward approach – dataset rewriting – showing that only substantial dataset rewriting efforts can alleviate this issue. Finally, we proposed CASCADE, a novel pretraining method, along with its model-compression version. Experiments demonstrate that CASCADE significantly outperforms baselines.

Despite its fundamental nature, our work has several limitations that may inspire future studies. First, we did not apply our training method to real-world datasets due to limited computational resources and lack of evaluation metrics. The qualitative pipeline in Appendix B may serve as a metric, but automatically selecting representative knowledge merits further study. Second, our study contains at most 3 modes. Future work could study an $n$-mode setting where $n > 3$ and compute the corresponding normalized log probabilities to verify CASCADE's advantage over dataset rewriting. Also, researchers can aim at finding other novel ideas to get better performances on cross-mode knowledge retrieval.

## Acknowledgement

RZ acknowledges helpful discussions with and support of (ordered lexicographically) Harkirat Behl, Sebastien Bubeck, Tong Chen, Yifang Chen, Simon Du, Beibin Li, Yuanzhi Li, Chen Liang, Zhihan Xiong, Dingli Yu, and Cyril Zhang.

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

# A   Additional related works

**Physics of language models.**   A line of closely related works are physics of language models (Allen-Zhu & Li, 2024a; Ye et al., 2024a;b; Allen-Zhu & Li, 2024b;c;d), which center around how language models learn and manipulate knowledge. Allen-Zhu & Li (2024a) demonstrates that transformer-based models like GPT (Radford et al., 2018; 2019; Brown et al., 2020; Achiam et al., 2023) can effectively learn and generate complex, recursive language structures from context-free grammars. In Ye et al. (2024a), the authors investigate how small language models solve grade-school math problems, distinguishing between memorization and genuine reasoning. Ye et al. (2024b) focuses on improving models' reasoning accuracy by incorporating"retry data" during pretraining stage. Allen-Zhu & Li (2024b) finds that knowledge augmentation during pretraining significantly improves the models' ability to extract and utilize knowledge, introduces novel probing techniques to understand this process, and suggests to enhance language model training with *data rewriting* and early introduction of question-answering tasks. Allen-Zhu & Li (2024c) explores the limitations of language models in executing basic knowledge manipulation tasks—retrieval, classification, comparison, and inverse search. It proposes methods like generating more Chain-of-Though (CoT, Wei et al. (2022)) data, employing retrieval augmented generation (RAG, Lewis et al. (2020)) and reversal training. Allen-Zhu & Li (2024d) presents a comprehensive study on the knowledge capacity scaling laws of language models, revealing that a 2bit/param capacity ratio is achievable across various architectures and training conditions, but is affected by factors such as training exposure, model architecture, quantization, sparsity, and the quality of training data.

**Spurious correlations.**   Spurious correlations represent a significant threat to the reliability and trustworthiness of NLP systems, as they can cause models to learn unintended shortcuts rather than the underlying task-relevant signals (Eisenstein, 2022; Wang et al., 2022). This issue has been widely studied in text classifications tasks. Joshi et al. (2022) examines spurious features through a causal lens, classifying them based on probability of necessity (PN) and probability of sufficiency (PS). They identify two categories: irrelevant features (low PN, low PS) and necessary features (high PN, low PS). Wu et al. (2022) introduce a data generation approach to mitigate spurious correlations by creating debiased versions of datasets. Bansal & Sharma (2023) estimate the causal effect of features on labels and regularize models to match this true effect, developing an automated augmentation method that improves performance on minority groups while maintaining overall accuracy. Lee et al. (2024) build a human-model interaction interface, allowing users to give descriptions about models' misconceptions about spurious correlations, ultimately improving the performance.

# B   Qualitative studies

## B.1   Setup

For qualitative studies, we use paragraphs from Wikipedia as test cases. We manually select a sentence from the original Wikipedia text, replace it with a [BLANK] along with its hint. We call this input `original`, and the selected sentence is called `answer`.

Next, we prompt GPT-4o using the template in Text Box 1, replacing {text} with `original`. This generates a story-style text called `altered`, which contains a corresponding [BLANK] with a hint.

We then separately prompt GPT-4o 100 times using the template in Text Box 2, replacing {text} with `original` and `altered`, respectively. To avoid API-side caching, we add ATTEMPT {i} to the beginning of each prompt. This generates responses $r_{\text{original}}^{(i)}$ and $r_{\text{altered}}^{(i)}$ for $1 \leqslant i \leqslant 100$.

Finally, we prompt GPT-4o using the template in Text Box 3, replacing {text} with type, {response} with $r_{\text{type}}^{(i)}$, and {answer} with `answer`, where type $\in$ {`original`,`altered`} and $1 \leqslant i \leqslant 100$. We extract accuracies from the judge outputs and average them.

```
You will help me rewrite a text into another style.
I will give you a text based on a fact from Wikipedia.
I left a blank, [BLANK], as well as its hint in the text.
Your task is to rewrite the text into a story, under the setting that a mother is telling a bedtime
    story to her kid.
Aside from the information in the original text, you should describe about the environment, the
    characters, and the plot.
The rewritten text should be coherent and consistent with the original text.
You must retain the blank and its hint in the rewritten text, for example, when the hint requires to
    output three items, you should include the hint in the rewritten text as well.

===== Text =====
{text}
```

Text Box 1: The input template for rewriting into a story style.

```
I will give you a text based on a fact.
I left a blank, [BLANK], as well as its hint in the text.
Please fill in the blank after you read the text.
You should provide the most appropriate information, as accurate as possible.

===== Text =====
{text}
```

Text Box 2: The input template for blank completion.

```
You are a judge to evaluate the response of the completion system.
I'll provide you a text with a blank, [BLANK].
Then, I'll provide you a response to fill in the blank, and its ground truth answer.
Please evaluate whether the response is correct or not, output a float number between 0 and 1 to
    represent the accuracy.
Identify each important aspects in the ground truth answer, and compare them with the response.
The floating number should be finally outputed in the following format:
```Accuracy
[ACCURACY]
```

===== Text =====
{text}

===== Response =====
{response}

===== Ground Truth =====
{answer}
```

Text Box 3: The input template for judging a completion response.

## B.2 Results

We present three examples in Tables 3 to 5. Detailed results are included in `scripts/eval/data.json` in the supplementary materials.

| original | |
|---|---|
| | Masayoshi Soken (born January 10, 1975) is a Japanese video game composer and sound editor who has worked for Square Enix since 1998. Soken is best known for being the lead composer and sound director of Final Fantasy XIV and its expansions and lead composer of Final Fantasy XVI.

Biography
[BLANK] (Hint: Write about Soken's born city, univeristy, and major.) He was exposed to music at a young age - his father was a professional trumpet player and his mother taught piano. Rather than pursue a career in science, he was hired as a sound editor and designer at Konami. At Konami, he worked on the arcade rail shooting game Evil Night. He joined Square in 1998 and his early work at the company primarily focused on sound effect design, rather than composition. He assisted Kenji Ito on sports video games such as Gekikuukan Pro Baseball: The End of the Century 1999. His debut as a composer came with the Japan-exclusive sports games Nichibeikan Pro Baseball: Final League, in which he was the sole composer, and World Fantasista with synthesizer programmer Takeharu Ishimoto. |
| answer | Born in La Paz, Baja California Sur, Mexico, Soken and his family later moved to Tokyo, where he attended the Tokyo University of Science, majoring in chemistry. |
| Example response | Masayoshi Soken was born in La Paz, Mexico, but moved to Japan at a young age. He attended the Tokyo University of Science, where he majored in chemistry. Despite his scientific studies, his passion for music ultimately led him to a career in video game composition. |
| Judge | [...]
All four aspects align with the ground truth, so the response accurately fills the blank with the correct information. Therefore, the accuracy score is:
```Accuracy
1.0
``` |
| altered | [...]
As Masayoshi grew, so did his love for music. He studied diligently, learning how different notes fit together like puzzle pieces to create beautiful symphonies. The city of [BLANK] (Hint: born city) whispered tales of his origins, and at [BLANK] (Hint: university), where he immersed himself in [BLANK] (Hint: major), Masayoshi's passion blossomed like a field of wildflowers under the summer sun.
[...] |
| Example response | [...]
The city of La Paz (Hint: born city) whispered tales of his origins, and at Tokyo College of Music (Hint: university), where he immersed himself in music composition (Hint: major), Masayoshi's passion blossomed like a field of wildflowers under the summer sun.
[...] |
| Judge | To evaluate the response, I'll compare each aspect of the ground truth with the provided filled text.

1. **Born City:**
- **Ground Truth:** La Paz, Baja California Sur, Mexico
- **Response:** La Paz
- **Evaluation:** Partial match for the city, but missing "Baja California Sur, Mexico."

2. **University:**
- **Ground Truth:** Tokyo University of Science
- **Response:** Tokyo College of Music
- **Evaluation:** Incorrect university.

3. **Major:**
- **Ground Truth:** Chemistry
- **Response:** Music composition
- **Evaluation:** Incorrect major.

Overall, the response contains significant inaccuracies, especially in the university and major fields. The born city partially matches. Based on the accuracy of these critical elements, I'll give an accuracy score.

```Accuracy
0.2
``` |

Table 3: Example 1: the average accuracies of the responses for the original input and altered input are 48.0% and 25.9%, respectively.

| original | |
|---|---|
| | Khalil Fong Tai Tung (14 July 1983 - 21 February 2025) was a Hong Kong-based American singer-songwriter and producer. Fong was notable for introducing a more sophisticated R\&B and soul sound to the Chinese music market, drawing inspiration from artists like Stevie Wonder, Musiq Soulchild and Earth, Wind \& Fire.

Early life and education
Khalil Fong was born on 14 July 1983 in Hawaii, United States, to an American-Chinese father, a drummer, and Hong Kong mother, Leung Yu-laam, an English language educator, who would later be Fong's lyricist under the pen name Rulan.

In 2005, Fong finally signed a recording contract with Warner Records. He then debuted with the release of his first album. [BLANK] (Hint: first four albums in chronological order, with names and years.) |
| answer | Soulboy (2005), This Love (2006), Wonderland (2007), Orange Moon (2008) |
| Example response | In 2005, Fong finally signed a recording contract with Warner Records. He then debuted with the release of his first album, "Soul Boy" (2005). His subsequent albums were "This Love" (2006), "Wonderland" (2007), and "Orange Moon" (2008). |
| Judge | [...]
Thus, the response's accuracy is very high despite the minor formatting variance for one album. The substantive content is correct.

```Accuracy
0.95
``` |
| altered | [...]
Now just imagine, four albums, each like a chapter in Khalil's tale, walking hand in hand down the path of time. [BLANK] (Hint: first four albums in chronological order, with names and years.) These albums were like magic spells, spreading beautiful music across the land and filling hearts with joy.
[...] |
| Example response | Khalil released his albums over a span of several years, each resonating with his unique sound. The first album was "SoulBoy" released in 2005, followed by "SoulBoy Lights Up" in 2006. His third album, "Wonderland," was launched in 2007, showcasing his evolving artistry. Finally, in 2008, Khalil released "Orange Moon," completing the magical quartet that enchanted listeners across the land. |
| Judge | To evaluate the response, let's compare each element with the ground truth:

1. **First Album:**
- Response: "SoulBoy" released in 2005
- Ground Truth: "Soulboy" released in 2005
- Evaluation: Correct (matches in both name and year)

2. **Second Album:**
- Response: "SoulBoy Lights Up" released in 2006
- Ground Truth: "This Love" released in 2006
- Evaluation: Incorrect (name does not match, though the year is correct)

3. **Third Album:**
- Response: "Wonderland" released in 2007
- Ground Truth: "Wonderland" released in 2007
- Evaluation: Correct (matches in both name and year)

4. **Fourth Album:** - Response: "Orange Moon" released in 2008
- Ground Truth: "Orange Moon" released in 2008
- Evaluation: Correct (matches in both name and year)

The response correctly identifies the first, third, and fourth albums in terms of both name and year, but it provides an incorrect name for the second album. Therefore, 3 out of 4 elements are correct.

```Accuracy
0.75
``` |

Table 4: Example 2: the average accuracies of the responses for the original input and altered input are 93.3% and 62.0%, respectively.

| original | |
|---|---|
| | Ted Chiang (born 1967) is an American science fiction writer. His work has won four Nebula awards, four Hugo awards, the John W. Campbell Award for Best New Writer, and six Locus awards. He has published the short story collections Stories of Your Life and Others (2002) and Exhalation: Stories (2019). His short story "Story of Your Life" was the basis of the film Arrival (2016). His father, Fu-pen Chiang, is a [BLANK] (Hint: occupation and organization). |
| answer | |
| | His father, Fu-pen Chiang, is a professor of mechanical engineering at Stony Brook University. |
| Example response | |
| | professor emeritus of mechanical engineering at Stony Brook University |
| Judge | |
| | [...]
Response analysis:
1. The occupation is expanded as "professor emeritus," which implies a retired professor with distinguished services, but still aligns with the role "professor."
2. The field of study "mechanical engineering" is identical in both the response and the ground truth.
3. The organization "Stony Brook University" is mentioned correctly in both.
[...]
```Accuracy
1.0
``` |
| altered | |
| | [...]
"Ted had a source of inspiration that was close to his heart – his father, Fu-pen Chiang, who was a [BLANK] (Hint: occupation and organization)." The mother paused, letting her child imagine what wondrous occupation Ted's father might have had, perhaps working in a fantastic organization that helped spark Ted's creativity.
[...] |
| Example response | |
| | engineer at the Jet Propulsion Laboratory |
| Judge | |
| | [...]
The ground truth specifies that Fu-pen Chiang is a professor of mechanical engineering, whereas the response identifies him as an engineer. This indicates a difference in professional designation and field of work.\n- The affiliation in the ground truth is Stony Brook University, which is different from the response's Jet Propulsion Laboratory.

Both the occupation and the affiliation in the response do not match those in the ground truth. Therefore, the accuracy of the response is quite low.

```Accuracy
0.0
``` |

Table 5: Example 3: the average accuracies of the responses for the original input and altered input are 78.3% and 28.5%, respectively.

## C  Justifications for CASCADE

### C.1  Explanation for knowledge capture

Here we justify that this cascading design of datasets ensures that each piece of knowledge is captured by *at least one* language model. Consider a piece of knowledge with length $L_{\text{knw}}$ that appears in position $(p, p + 1, \ldots, p + L_{\text{knw}} - 1)$ of $\mathcal{D}$. Then for each $3 \leqslant m \leqslant M$, we identify the training sequences which contain the knowledge across the halfway point (i.e., sequences that have this knowledge in both the hint and completion parts):

$$
\begin{cases}
i \cdot 2^{m-1} \leqslant p, & \text{① Training sequence starts before } \textit{knowledge}; \\
i \cdot 2^{m-1} + 2^{m-1} > p, & \text{② Hint contains } \textit{knowledge}; \\
i \cdot 2^{m-1} + 2^{m-1} \leqslant p + L_{\text{knw}} - 1, & \text{③ Completion contains } \textit{knowledge}.
\end{cases}
$$

With all requirements combined, we can solve for $i$:

$$
\frac{p}{2^{m-1}} - 1 < i \leqslant \min\left\{\frac{p}{2^{m-1}}, \frac{p + L_{\text{knw}} - 1}{2^{m-1}} - 1\right\}. \tag{1}
$$

If $L_{\text{knw}} \geqslant 2^{m-1} + 1 > L_{\text{ctx}}^m/2$, then there is a unique solution $i = \lfloor p/2^{m-1} \rfloor$. Here requirement ① is optional, because without it means the training sequence does not have mode in the hint part, which is helpful for knowledge completion. Thus, for all $m \leqslant 1 + \lfloor \log(L_{\text{knw}} - 1) \rfloor$, this piece of knowledge occupies half of a training sequence in $\mathcal{D}_m$ and is therefore captured by model $\mathcal{M}_m$.

### C.2  Theoretical time complexity analysis for CASCADE

In self-attention (Waswani et al., 2017), processing a batch of $B$ training sequences with length $L_{\text{ctx}}$ takes $\Theta(B(L_{\text{ctx}})^2)$ time.

**Training/Evaluation.**  Suppose we use the efficient evaluation method in Section 5.1.2, then training and evaluation are essentially the same (except for a backward pass). The time complexity is

$$
\sum_{m=3}^{M} \Theta(B_m(L_{\text{ctx}}^{(m)})^2) = \sum_{m=3}^{M} \Theta(2BL_{\text{ctx}}L_{\text{ctx}}^{(m)}) = \sum_{m=3}^{M} \Theta(2BL_{\text{ctx}}2^m) = \Theta(B(L_{\text{ctx}})^2)
$$

as we recall $M = \log_2(2\overline{L_{\text{knw}}}) = \log_2 L_{\text{ctx}}$.

**Inference.**  Suppose batch size $B = 1$ in inference. To generate a single sequence using the original method, the time complexity is

$$
\sum_{p=1}^{L_{\text{ctx}}} \Theta(p^2) = \Theta((L_{\text{ctx}})^3).
$$

For CASCADE, the time complexity is

$$
\sum_{p=1}^{L_{\text{ctx}}} \sum_{m=3}^{M} \Theta(\min\{p^2, (L_{\text{ctx}}^{(m)}/2)^2\}) \leqslant \sum_{p=1}^{L_{\text{ctx}}} \sum_{m=3}^{M} \Theta((L_{\text{ctx}}^{(m)}/2)^2) = \sum_{p=1}^{L_{\text{ctx}}} \sum_{m=3}^{M} \Theta(4^m) = \Theta((L_{\text{ctx}})^3)
$$

as we recall $M = \log_2 L_{\text{ctx}}$.

Therefore, from a theoretical perspective, CASCADE does not introduce much time overhead.

# D Experiment details for the quantitative experiments

## D.1 Datasets

There are 473992236 tokens in $\mathcal{F}_{ts}$, 484159419 tokens in $\mathcal{F}_{wiki}$, and 484626190 tokens in $\mathcal{F}_{code}$ (mentioned in Appendix D.4.2). When constructing $\mathcal{D}_{ts}$, $\mathcal{D}_{wiki}$ and $\mathcal{D}_{code}$, regardless of the random seed, the data are arranged in a fixed order such that all types ($f_{wiki}\,k_{wiki}$, $f_{wiki}\,k_{ts}$, $f_{wiki}\,k_{code}$, $f_{ts}\,k_{ts}$, $f_{ts}\,k_{wiki}$, $f_{ts}\,k_{code}$, $f_{code}\,k_{code}$, $f_{code}\,k_{wiki}$, $f_{code}\,k_{ts}$) of data are approximately uniformly distributed. All the datasets use the same *dataset seed* 42 which is independent of the random seeds in training, to ensure that all the datasets are the same across different runs. The hyperparameters for dataset construction are listed in Table 6.

| Hyperparameter | Value |
|---|---|
| Random token sequence | |
| - Length range | $[8, 512]$ |
| - Different sequences per dataset | 32 |
| - $N_{occ}$ | 8192 |
| - $N_{occ}^{\times}$ | $\{0\} \cup \{2^i \mid 1 \leqslant i \leqslant 13\}$ |
| Dataset seed | $\{42\}$ |

Table 6: Hyperparameters of datasets

## D.2 Models

We use a customized phi-1 (162M) model specified in Table 7, where the original architecture is introduced in Gunasekar et al. (2023). The value of n_positions corresponds to the sequence lengths in training (see Table 8).

| Specification | Value |
|---|---|
| Type | mixformer-sequential |
| Architecture | |
| - block_cls | parallel |
| - mixer | |
|   - mixer_cls | mha |
|   - dropout | 0.1 |
| - mlp_cls | fused_mlp |
| Total parameters | 162m |
| - vocab_size | 50304 |
| - n_positions | $\{2^m \mid 3 \leqslant m \leqslant 10\}$ |
| - n_embd | 768 |
| - n_layer | 12 |
| - n_head | 12 |
| - rotary_dim | 32 |
| resid_pdrop | 0.1 |

Table 7: Model specification of our customized phi-1 (162M).

## D.3 Training

We use the set of hyperparameters in Table 8. For the experiments of *ablation on practical running time* in Section 5.3, we use 16 epochs, while for all the other experiments, we use 2 epochs. For the experiments for dataset rewriting in Section 4.2, we use sequence lengths of 1024, while for the experiments of cascading datasets in Section 5.3, $m$ can vary between $3, 4, \ldots, 10$.

## D.4 More results

Here we list results not being able to be presented in the main text due to page limit.

| Hyperparameter | Value |
|---|---|
| Number of epochs | $\{2, 16\}$ |
| Train batch size | 1024 |
| Optimizer | AdamW |
| - Gradient clipping norm | 1.0 |
| - $\beta_1, \beta_2$ | $0.9, 0.95$ |
| - $\epsilon$ | $1 \times 10^{-7}$ |
| - Weight decay | 0.1 |
| Learning rate scheduler | WarmupDecayLR |
| - Warmup min lr | $1 \times 10^{-7}$ |
| - Warmup max lr | $1 \times 10^{-4}$ |
| - Warmup steps | 500 |
| - Warmup type | Linear |
| Precision | fp16 (initial scale power: 12) |
| Sequence length | $\{2^m \mid 3 \leqslant m \leqslant 10\}$ |
| Random seed | $\{42, 142857, 2225393, 20000308, 2018011309\}$ |

Table 8: Hyperparameters of the quantitative experiments

### D.4.1 Evaluating each context length in CASCADE

To see how each context length $L_{\text{ctx}}^{(m)}$ contributes to the performance of $\mathcal{M}_\theta$, we conducted 3 extra evaluations: (1) Evaluation with exactly one context length (Table 9), (2) Evaluation without exactly one context length (Table 10), and (3) Visualization of the contributions of each context length at each sequence position (Figure 7).

| Context Length | $f_{\text{ts}}\, q_{\text{ts}}$ | $f_{\text{wiki}}\, q_{\text{wiki}}$ | $f_{\text{ts}}\, q_{\text{wiki}}$ | $f_{\text{wiki}}\, q_{\text{ts}}$ |
|---|---|---|---|---|
| 8 | $-5.00 \times 10^{-1}$ | $-4.98 \times 10^{-1}$ | $-4.98 \times 10^{-1}$ | $-5.00 \times 10^{-1}$ |
| 16 | $-3.21 \times 10^{-5}$ | $-2.64 \times 10^{-5}$ | $-4.11 \times 10^{-5}$ | $-5.56 \times 10^{-5}$ |
| 32 | $-6.74 \times 10^{-7}$ | $-6.65 \times 10^{-7}$ | $-1.01 \times 10^{-5}$ | $-1.82 \times 10^{-5}$ |
| 64 | $-4.94 \times 10^{-7}$ | $-5.08 \times 10^{-7}$ | $-1.35 \times 10^{-5}$ | $-1.72 \times 10^{-5}$ |
| 128 | $-4.35 \times 10^{-7}$ | $-4.63 \times 10^{-7}$ | $-1.27 \times 10^{-5}$ | $-1.78 \times 10^{-5}$ |
| 256 | $-4.07 \times 10^{-7}$ | $-4.91 \times 10^{-7}$ | $-1.39 \times 10^{-5}$ | $-1.55 \times 10^{-5}$ |
| 512 | $-3.85 \times 10^{-7}$ | $-5.33 \times 10^{-7}$ | $-1.68 \times 10^{-5}$ | $-1.62 \times 10^{-5}$ |
| 1024 | $-3.68 \times 10^{-7}$ | $-5.58 \times 10^{-7}$ | $-2.00 \times 10^{-5}$ | $-2.17 \times 10^{-5}$ |

Table 9: Normalized log probabilities of the model trained using CASCADE with overlapping sequences, evaluated using individual fixed context lengths. The values are averaged over 5 random seeds.

| Context Length | $f_{\text{ts}}\, q_{\text{ts}}$ | $f_{\text{wiki}}\, q_{\text{wiki}}$ | $f_{\text{ts}}\, q_{\text{wiki}}$ | $f_{\text{wiki}}\, q_{\text{ts}}$ |
|---|---|---|---|---|
| 8 | $-3.27 \times 10^{-7}$ | $-3.46 \times 10^{-7}$ | $-3.71 \times 10^{-6}$ | $\mathbf{-5.05 \times 10^{-6}}$ |
| 16 | $-3.42 \times 10^{-7}$ | $-3.68 \times 10^{-7}$ | $-5.37 \times 10^{-6}$ | $-8.30 \times 10^{-6}$ |
| 32 | $-3.35 \times 10^{-7}$ | $-3.60 \times 10^{-7}$ | $-4.08 \times 10^{-6}$ | $-5.59 \times 10^{-6}$ |
| 64 | $\mathbf{-3.23 \times 10^{-7}}$ | $-3.45 \times 10^{-7}$ | $-3.85 \times 10^{-6}$ | $\mathbf{-4.95 \times 10^{-6}}$ |
| 128 | $\mathbf{-3.22 \times 10^{-7}}$ | $\mathbf{-3.43 \times 10^{-7}}$ | $\mathbf{-3.66 \times 10^{-6}}$ | $\mathbf{-4.89 \times 10^{-6}}$ |
| 256 | $\mathbf{-3.24 \times 10^{-7}}$ | $\mathbf{-3.42 \times 10^{-7}}$ | $\mathbf{-3.68 \times 10^{-6}}$ | $\mathbf{-5.04 \times 10^{-6}}$ |
| 512 | $-3.28 \times 10^{-7}$ | $\mathbf{-3.44 \times 10^{-7}}$ | $\mathbf{-3.67 \times 10^{-6}}$ | $\mathbf{-5.02 \times 10^{-6}}$ |
| 1024 | $-3.37 \times 10^{-7}$ | $-3.50 \times 10^{-7}$ | $\mathbf{-3.50 \times 10^{-6}}$ | $-5.12 \times 10^{-6}$ |

Table 10: Normalized log probabilities of the model trained using CASCADE with overlapping sequences, evaluated *without* specific context lengths. The values are averaged over 5 random seeds. Values better than those in Table 2 (CASCADE Overlap) are in bold red text.

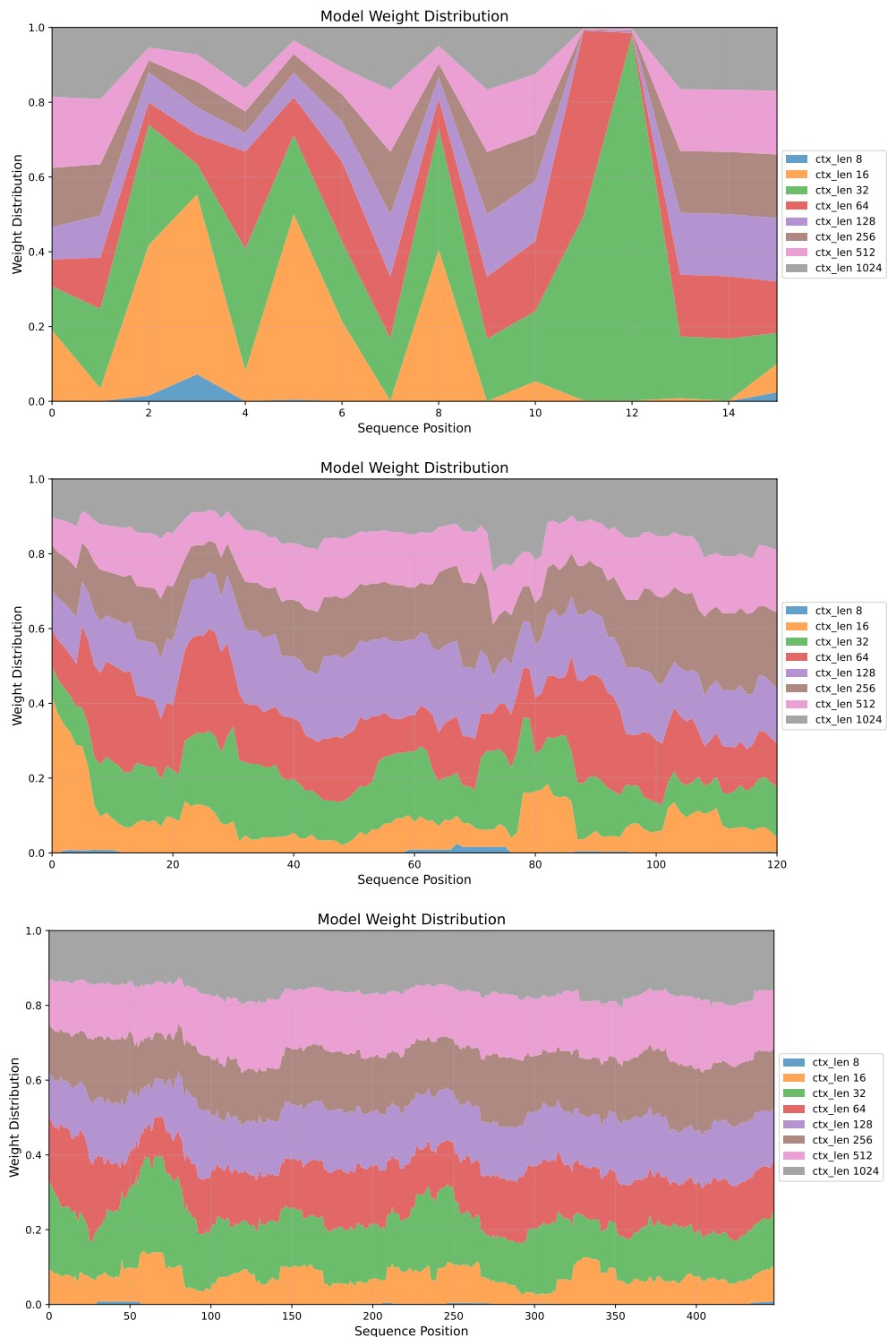

Figure 7: Weight distribution over models for different positions in the completion part.

### D.4.2 Extension to 3 modes and bigger models

To show the generalization capability of CASCADE, we conducted additional experiments on 3 modes. We used Python code from the `bigcode/the-stack` dataset (Kocetkov et al., 2022) as a new mode – code. In Table 11, we present results for dataset rewriting with $r = \infty$ (direct training), $r = 1$ (full rewriting), $r = 1$ but *without* $f_{\text{wiki}}\ q_{\text{ts}}$ in the training set, and CASCADE using the customized phi-1 (162M) specified in Appendix D.2. Further, we use the model architecture of phi-1-small (350M) (Gunasekar et al., 2023), which roughly doubles the size of our customized phi-1 (162M), to see how baselines scale. The results are shown in Table 12.

| | $f_{\text{ts}}\, q_{\text{ts}}$ | $f_{\text{wiki}}\, q_{\text{wiki}}$ | $f_{\text{code}}\, q_{\text{code}}$ | $f_{\text{ts}}\, q_{\text{wiki}}$ | $f_{\text{ts}}\, q_{\text{code}}$ | $f_{\text{wiki}}\, q_{\text{ts}}$ | $f_{\text{wiki}}\, q_{\text{code}}$ | $f_{\text{code}}\, q_{\text{ts}}$ | $f_{\text{code}}\, q_{\text{wiki}}$ |
|---|---|---|---|---|---|---|---|---|---|
| Direct training ($r=\infty$) | $-7.74 \times 10^{-6}$ | $-8.39 \times 10^{-6}$ | $-1.75 \times 10^{-5}$ | $-0.215$ | $-0.755$ | $-0.368$ | $-8.79 \times 10^{-2}$ | $-0.724$ | $-2.80 \times 10^{-2}$ |
| Full rewriting ($r=1$) | $-2.62 \times 10^{-5}$ | $-2.78 \times 10^{-5}$ | $-3.10 \times 10^{-5}$ | $-6.89 \times 10^{-5}$ | $-1.33 \times 10^{-4}$ | $-3.46 \times 10^{-4}$ | $-8.45 \times 10^{-5}$ | $-1.51 \times 10^{-4}$ | $-1.84 \times 10^{-4}$ |
| $r=1$ without $f_{\text{wiki}}\, q_{\text{ts}}$ | $-2.87 \times 10^{-5}$ | $-6.37 \times 10^{-6}$ | $-2.52 \times 10^{-5}$ | $-2.02 \times 10^{-4}$ | $-7.71 \times 10^{-5}$ | $\mathbf{-4.67 \times 10^{-3}}$ | $-1.37 \times 10^{-4}$ | $-1.52 \times 10^{-4}$ | $-6.32 \times 10^{-5}$ |
| CASCADE | $-1.97 \times 10^{-7}$ | $-1.69 \times 10^{-7}$ | $-4.24 \times 10^{-7}$ | $-1.08 \times 10^{-6}$ | $-7.01 \times 10^{-5}$ | $-8.01 \times 10^{-5}$ | $-9.42 \times 10^{-5}$ | $-6.69 \times 10^{-5}$ | $-2.82 \times 10^{-7}$ |

Table 11: Normalized log probabilities when there are 3 modes. In the experiment of full rewriting without $f_{\text{wiki}}\, q_{\text{ts}}$, the evaluation result of $f_{\text{wiki}}\, q_{\text{ts}}$ is marked in bold blue text. That result is one order worse than the other cross-mode results.

| | $f_{\text{ts}}\, q_{\text{ts}}$ | $f_{\text{wiki}}\, q_{\text{wiki}}$ | $f_{\text{code}}\, q_{\text{code}}$ | $f_{\text{ts}}\, q_{\text{wiki}}$ | $f_{\text{ts}}\, q_{\text{code}}$ | $f_{\text{wiki}}\, q_{\text{ts}}$ | $f_{\text{wiki}}\, q_{\text{code}}$ | $f_{\text{code}}\, q_{\text{ts}}$ | $f_{\text{code}}\, q_{\text{wiki}}$ |
|---|---|---|---|---|---|---|---|---|---|
| Direct training ($r=\infty$) | $-6.73 \times 10^{-7}$ | $-9.03 \times 10^{-7}$ | $-6.25 \times 10^{-7}$ | $-4.19 \times 10^{-2}$ | $-0.277$ | $-5.54 \times 10^{-2}$ | $-1.39 \times 10^{-2}$ | $-0.172$ | $-3.80 \times 10^{-3}$ |
| Full rewriting ($r=1$) | $-5.68 \times 10^{-6}$ | $-5.80 \times 10^{-6}$ | $-6.25 \times 10^{-6}$ | $-2.21 \times 10^{-5}$ | $-2.89 \times 10^{-5}$ | $-6.16 \times 10^{-5}$ | $-5.51 \times 10^{-5}$ | $-3.12 \times 10^{-5}$ | $-1.65 \times 10^{-5}$ |

Table 12: Normalized log probabilities when using phi-1-small (350M) (Gunasekar et al., 2023).

