# OpenReview forum: "CASCADE Your Datasets for Cross-Mode Knowledge Retrieval of Language Models"
_colmweb.org/COLM/2025/Conference — COLM 2025_

### Official Review · Reviewer_cKFW · 2025-05-08

**Rating:** 7
**Confidence:** 4
**Ethics Flag:** 1

**Summary:**

This paper investigates the cross-mode knowledge retrieval capability of language models in a cross-mode setting. The authors first quantify the analysis with a knowledge memorization setting, with knowledge pieces defined as sequences sampled randomly with log probability constraints. In the experiments, two formats of pre-training data are considered as modes, and data rewriting is treated as the baseline. Upon that, the authors propose CASCADE, which considers different context lengths with weighted losses during pre-training. The authors demonstrate significant performance (in terms of the normalized log probabilities of the knowledge pieces) improvement on knowledge retrieval capability compared to the baselines.

**Questions To Authors:**

1. In section 3.1, the authors define the term “knowledge” used in the paper. However, “knowledge” can be an overshadowing term. Random token sequences cannot always be considered as knowledge since tokens are connected semantically. For example, the knowledge in the natural language case can be the relations among certain entities [4]. It would be good if the authors can add discussion on a potential direction where the knowledge constructing is with N-grams, which can potentially lead to more natural synthesis.
It is worth noting that “perfectly generate the whole sequences” does not rigorously mean knowledge memorization. Although this does not influence my rating, [5] would be worth reading in this direction. Generally speaking. Sequence-level exclusiveness may not be enough in some cases, where the N-gram overlaps should be considered. In addition, the sequence-level performance is also relevant to the n-gram (finer-granularity) behavior [6].

2. There is a line of work on the influence of model performance with JSON formatting (see: https://blog.dottxt.co/index.html). How is the JSON formatting related to the mode defined here?

[4] Synthetic continued pretraining

[5] Language Models May Verbatim Complete Text They Were Not Explicitly Trained On

[6] Potential related reading for varying context length: Fractal Patterns May Illuminate the Success of Next-Token Prediction

**Reasons To Accept:**

1. The idea is interesting. The way to quantify the mode difference can be generalized

2. This paper is well-structured and well-written. The authors clearly show the contribution with a series of findings from qualitative and quantitative analysis, to data and algorithmic design.

3. The algorithm proposed is insightful and neat. Recent papers in retrieval also consider incorporating this multi-granularity awareness.

4. The additional related work section is well-written and clearly supports the background of the proposed settings and methods.

**Reasons To Reject:**

1.  The illustrative figure can be made better (e.g., highlighting the format differences and explaining more about the red marks).

2. Although the following does not influence the novelty of the idea, the idea of packing different scales is closely related to the multi-granularity idea in the relevant knowledge retrieval community, e.g., [1][2][3]. It would be good if the authors could utilize these works to help motivate the proposed method in Section 5.1.

3. The experiment suite is slight limited with two datasets. Two modes are studied in this work. Although pretraining experiments can be costly, it would be appreciated if the authors can show some generalizability to other modes, e.g., some intermediate states between the current ones.

[1] Matryoshka Representation Learning

[2] Sub-Sentence Encoder: Contrastive Learning of Propositional Semantic Representations

[3] Dense X Retrieval: What Retrieval Granularity Should We Use?

---

> ### Author Response · Authors · 2025-06-02
> **Reply to Reviewer cKFW**
>
> ### Reasons To Reject:
>
> **[R1]** The illustrative figure can be made better (e.g., highlighting the format differences and explaining more about the red marks).
>
> [Reply] Thanks for your suggestion!
>
> **[R2]** Although the following does not influence the novelty of the idea, the idea of packing different scales is closely related to the multi-granularity idea in the relevant knowledge retrieval community, e.g., [1][2][3]. It would be good if the authors could utilize these works to help motivate the proposed method in Section 5.1.
>
> [Reply] This is an interesting line of related work. Matryoshka representation learning handles cascading feature series for improved classification, while sub-sentence encoders and dense X retrieval are more closely related as both process textual information using different text units. We will discuss these works (and other related ones) in greater detail in Section 5.1 to provide better motivation.
>
> **[R3]** The experiment suite is slight limited with two datasets. Two modes are studied in this work. Although pretraining experiments can be costly, it would be appreciated if the authors can show some generalizability to other modes, e.g., some intermediate states between the current ones.
>
> [Reply] It is hard to quantify "intermediate" modes, but we conducted experiments with another mode -- code. Please refer to the standalone reply for generalizing into $3$ modes.
>
> [1] Matryoshka Representation Learning
>
> [2] Sub-Sentence Encoder: Contrastive Learning of Propositional Semantic Representations
>
> [3] Dense X Retrieval: What Retrieval Granularity Should We Use?
>
> ### Questions To Authors:
>
> **[Q1]** In section 3.1, the authors define the term “knowledge” used in the paper. However, “knowledge” can be an overshadowing term. Random token sequences cannot always be considered as knowledge since tokens are connected semantically. For example, the knowledge in the natural language case can be the relations among certain entities [4]. It would be good if the authors can add discussion on a potential direction where the knowledge constructing is with N-grams, which can potentially lead to more natural synthesis. It is worth noting that “perfectly generate the whole sequences” does not rigorously mean knowledge memorization. Although this does not influence my rating, [5] would be worth reading in this direction. Generally speaking. Sequence-level exclusiveness may not be enough in some cases, where the N-gram overlaps should be considered. In addition, the sequence-level performance is also relevant to the n-gram (finer-granularity) behavior [6].
>
> [Reply] Thank you for this valuable question! We chose random token sequences to avoid interference between knowledge and mode (as stated in Section 3.1), ensuring the quantitative study is well-defined. This approach prevents knowledge from becoming entangled with modes and each other, maintaining their independence and eliminating ambiguity in verbatim completion. Since each random token has $8$ choices, there are at most $8^n$ different $n$-grams. The expected length of each knowledge is roughly $256$, and there are $64$ total knowledge instances, yielding $256 \times 64 = 16384 = \frac{1}{2} \times 8^5$ total $n$-grams. When considering $n$-grams for $n > 5$, there is a very high probability that our random token knowledge contains no common $n$-grams, further confirming their independence.
>
> Meanwhile, one difficulty of using natural language knowledge is precisely that of defining knowledge independence. With $n$-grams, sentences with completely different $n$-gram patterns can convey identical meanings, e.g., "Fierce blizzard buried tiny hamlet, forcing school closures." versus "Intense snowstorm shut down village classrooms." This becomes highly probable when modes employ drastically different word styles. Consequently, querying such knowledge could yield unexpectedly strong performance because **it has already been rewritten**.
>
> We leave this important question as a future direction, while in this work we focus on the conceptually simpler task of memorizing independent random token sequences.
>
> **[Q2]** There is a line of work on the influence of model performance with JSON formatting (see: https://blog.dottxt.co/index.html). How is the JSON formatting related to the mode defined here?
>
> [Reply] Yes, this is a related problem. One of our initial motivations stems from observing that language model accuracy drops when querying knowledge originally in HTML format using JSON format instead. This issue also occurs when using GPT-4o, requesting structured outputs for easier parsing reduces accuracy compared to free generation.
>
> [4] Synthetic continued pretraining
>
> [5] Language Models May Verbatim Complete Text They Were Not Explicitly Trained On
>
> [6] Potential related reading for varying context length: Fractal Patterns May Illuminate the Success of Next-Token Prediction

---

> > ### Comment · Reviewer_cKFW · 2025-06-03
> >
> > Thanks so much for your detailed reply. I will keep my score as it is.

---

> > > ### Author Response · Authors · 2025-06-04
> > >
> > > Thank you for your support!

---

### Official Review · Reviewer_KQtE · 2025-05-09

**Rating:** 6
**Confidence:** 2
**Ethics Flag:** 1

**Summary:**

This paper explores the limitations of parametric knowledge in language models. The underlying hypothesis is that an LLM's ability to retrieve parametric knowledge is impacted by the form in which the request is made for that parametric knowledge. Even if an LLM is trained on multiple forms (such as storytelling and wikipedia), it will be unsuccessful at retrieving one type of knowledge if the other format is used to request the information

The paper explores two approaches to dealing with this failing of the LLM. The first is to use dataset rewriting. It appears that the intention is that dataset rewriting is a manual process where one dataset is rewritten to take on the form or style of another (such as wikipedia being rewritten as stories). The second is CASCADE, the proposed approach. In CASCADE, the two datasets are combined such that a chunk of one form overwrites a chunk is the other domain. The name CASCADE appears to come from the fact that multiple models are trained with different context lengths. The model with the longest context observes the entire chunk. Models trained on smaller contexts may see a mixture of the two forms or just one form. Model contexts range from 8 to 1024, each doubling in size. All models are trained to predict the second half of the sequence. The models of different context lengths are ensembled to produce valid probability distributions over tokens.

The two approaches are compared on sequences where the form is tiny stories and the queries are answered from wikipedia or vice-versa. These are compared to the performance where the form is held constant. Normalized log probabilities are reported. According to the authors, a better approaches will have more similar log probabilities. Using CASCADE, the non-overlap approach has similar results to results to rewriting (same orders of magnitude), but the overlapping approached have smaller differences in normalized log probabilities, although the normalized log probabilities are always smaller with the forms agree between the sequence f and the query q.

In general the problem of utilizing parametric knowledge for generated output in different forms is an interesting problem. Determining approaches to successfully address the weakness exposed by this work is important. Given the related work (which appears predominantly in the appendix), this is the first effort to explore this particular problem. However, the clarity of the paper makes the contribution of the work somewhat difficult to ascertain. In particular, the many crucial details appear in the appendix. The contribution cannot be understood without it. In addition, the baseline approach of dataset rewriting is dismissed as too expensive, but it appears the authors believe that this must be a manual effort. It is unclear if an LLM can be prompted to do the necessary re-writing.

Another weaknesses comes from the way the results are described. First it is unclear if differences or absolute values are more important when assessing performance. From Section 4.3 one would think it was differences, but from Section 5.3, it may be that best approaches will have the smaller normalized log probability of the cross-mode settings. This is unclear. Given that according to the authors, this is a new problem and a new evaluation, the measure used to present results as well as the way to assess if one is improving given that measure needs to be explained.

In addition it is unclear precisely how the training data in CASCADE is ensemble. Much more precision is put into describing the size of the data chunking then how two chunks are chosen to be combined. For instance can a random wikipedia chunk be combined with a random tiny story chunk or does there need to be some topical relation. How is the insertion point for the overwritten text chosen and how much text is overwritten? These are important points for anyone that attempts to replicate this work.

Other weaknesses include
- Too many key points appear in the appendix. At the very least the topics of other related works should appear in the main paper. Most of the remarks in Section 5.3 refer to the appendix indicating that the conclusions drawn were not presented in the main paper.
- Section 4.1 needs to indicate that the authors believe that rewriting the dataset is a manual process. A reader is likely to assume that an LLM will be used to rewrite. It is not until 4.3 that the reader is informed otherwise.
- Part of the challenge with the paper is that generic examples Figure 6 are used rather than filling in the block with actual text so that the reader is given an idea of what the dataset looks like

**Questions To Authors:**

Q1 Could you provide an explanation of what the reader should look for when presented with log-probabilities?
Q2 Is there a reason for dividing the results among 4 tables rather than presenting a single table so that all variants can be easily compared?
Q3 Please provide more details on how the training data is constructed.
Q4 Do you think this approach would work when training a model to handle code switching? Would that be a viable alternative use case to cross-mode retrieval?

**Reasons To Accept:**

+ The paper describes a new problem of LLMs -- to able to utilize parametric knowledge when the form of the generated text changes
+ While it is unclear precisely how costly re-writing a data set is, the proposed approach is definitely less costly, assuming that it is formed by randomly splicing two datasets together. Having easy access to training data is benefinical

**Reasons To Reject:**

- The paper lacks sufficient details to stand on its own without the appendix
- The measure used to ascertain whether the proposed approach is effective is not sufficiently defined
- While the paper claims to explore the problem from a qualitative and quantitative perspective, the main paper focuses on the quantitative results. The qualitative results only appear in the appendix.
- Unclear if the approach could handle more than 2 modes, which is a limitation to the generalizability of the approach

---

> ### Author Response · Authors · 2025-06-02
> **Reply to Reviewer KQtE (3/3)**
>
> ### Questions To Authors:
>
> **[Q1]** Could you provide an explanation of what the reader should look for when presented with log-probabilities?
>
> [Reply] Log-probability is a standard training and evaluation loss that should be maximized, i.e., as close to $0$ as possible. Let $\log p$ denote the average log-probability; then $\exp(L \cdot \log p)$ represents the probability of generating the desired sequence. When $\log p$ approaches $0$, $\exp(L \cdot \log p)$ approaches $1$, indicating the LLM is nearly deterministic (confident) in its prediction.
>
> **[Q2]** Is there a reason for dividing the results among 4 tables rather than presenting a single table so that all variants can be easily compared?
>
> [Reply] This is just because we want to present the results for each of the methods. It is a good idea to merge them together.
>
> **[Q3]** Please provide more details on how the training data is constructed.
>
> [Reply] We have already replied in W4.
>
> **[Q4]** Do you think this approach would work when training a model to handle code switching? Would that be a viable alternative use case to cross-mode retrieval?
>
> [Reply] Code switching has important applications in changing tone, genre, or domain-specific language. For example, "Summarize this legal paragraph in layman's terms, then rephrase it for a medical audience." While this appears related, a fundamental difference exists. Our work targets the **pretraining** stage, teaching the LLM knowledge for future retrieval. Code switching primarily focuses on **in-context instruction following** (typically at the **posttraining** stage), where knowledge is provided in the prompt and **the LLM doesn't need to retrieve it**. Therefore, CASCADE is not well-suited for code switching applications.

---

> ### Author Response · Authors · 2025-06-02
> **Reply to Reviewer KQtE (2/3)**
>
> ### Reasons To Reject:
>
> **[R1]** The paper lacks sufficient details to stand on its own with the appendix
>
> **[R2]** The measure used to ascertain whether the proposed approach is effective is not sufficiently defined
>
> [Reply] For the quantitative study, the performance measure is the normalized log probabilities (using absolute values rather than differences), which is well-defined for our case of random token sequences. For the qualitative study, we use another LLM as a judge to evaluate knowledge accuracy. There doesn't seem to be any accurate (not sample based) rule-based method to quantify natural language knowledge memorization given multiple representations of a knowledge.
>
> **[R3]** While the paper claims to explore the problem from a qualitative and quantitative perspective, the main paper focuses on the quantitative results. The qualitative results only appear in the appendix.
>
> [Reply] We reply to R1 and R3 together: this is mainly due to page limits and we are working on to make the presentation better.
>
> **[R4]** Unclear if the approach could handle more than 2 modes, which is a limitation to the generalizability of the approach
>
> [Reply] To address your concern, we conducted additional experiments on $n=3$ modes to show the efficacy of CASCADE, please read the standalone reply.

---

> ### Author Response · Authors · 2025-06-02
> **Reply to Reviewer KQtE (1/3)**
>
> ### Weaknesses
>
> **[W1]** However, the clarity of the paper makes the contribution of the work somewhat difficult to ascertain. In particular, the many crucial details appear in the appendix. The contribution cannot be understood without it.
>
> [Reply] Thanks for this concern. We've tried our best to present most results in main text, including the performance of CASCADE and all baselines. We will merge the table to save more space and present more results.
>
> **[W2]** In addition, the baseline approach of dataset rewriting is dismissed as too expensive, but it appears the authors believe that this must be a manual effort. It is unclear if an LLM can be prompted to do the necessary re-writing.
>
> [Reply] Dataset rewriting requires first identifying all the knowledge, then rewriting them. While LLMs can handle the rewriting step, the **identification** step requires significant human effort -- the trivial approach of rewriting the entire dataset is prohibitively expensive. Additionally, rewriting prompts need case-by-case design, which also demands substantial human effort. Our newly added experiments (those without $f_\text{wiki} q_\text{ts}$) support the hypothesis that when there are $n$ modes, all $n(n-1)/2$ pairs need rewriting. Therefore, even with capable LLMs, the cost remains too high.
>
> **[W3]** Another weaknesses comes from the way the results are described. First it is unclear if differences or absolute values are more important when assessing performance. From Section 4.3 one would think it was differences, but from Section 5.3, it may be that best approaches will have the smaller normalized log probability of the cross-mode settings. This is unclear. Given that according to the authors, this is a new problem and a new evaluation, the measure used to present results as well as the way to assess if one is improving given that measure needs to be explained.
>
> [Reply] We will improve our statements and we clarify it here: **absolute** values are more important, because during generation, the log probability itself directly affects the accuracy. Value differences were intended to show the desired (ideal) target performance for cross-mode knowledge retrieval.
>
> **[W4]** In addition it is unclear precisely how the training data in CASCADE is ensemble. Much more precision is put into describing the size of the data chunking then how two chunks are chosen to be combined. For instance can a random wikipedia chunk be combined with a random tiny story chunk or does there need to be some topical relation. How is the insertion point for the overwritten text chosen and how much text is overwritten? These are important points for anyone that attempts to replicate this work.
>
> [Reply] Thanks for raising this question. We create separate datasets for each mode and use a composite dataset to ensemble them. The `__getitem__()` method in the composite dataset maps an index to an index in a specific dataset, so **each training sequence contains data from only a single mode**. However, a training batch may contain data from several modes. Both the training sequences that receive knowledge insertions and the insertion points are uniformly random, as described in Section 3.2. To clarify: suppose there are $N$ training sequences in the Wikipedia dataset, and we want to insert $K N_{occ}$ knowledge instances, with $N_{occ}$ occurrences for each of the $K$ knowledge pieces. We call `idxs=rng.choice(N, size=K*N_occ, replace=True)` to obtain the indices for training sequences that will receive knowledge insertions. Since the returned order is uniformly random in the permutation space, we insert knowledge $i$ to the indices in `idxs[i*N_occ:(i+1)*N_occ]`. When inserting knowledge of length $l$ into a training sequence of length $L$, we call `p=rng.choice(range(0, L-l+1))` to determine the starting position. Finally, we overwrite positions `[p:p+l]`. The implementation is included in the supplementary files -- please refer to `project/datasets/lm/lm_dataset.py`.
>
> **[W5]** Too many key points appear in the appendix. At the very least the topics of other related topics should appear in the main paper. Most of the remarks in Section 5.3 refer to the appendix indicating that the conclusions drawn were not presented in the main paper.
>
> [Reply] We will arrange the presentation better.
>
> **[W6]** Section 4.1 needs to indicate that the authors believe that rewriting the dataset is a manual process....
>
> [Reply] We will edit accordingly.
>
> **[W6]** Part of the challenge with the paper is that generic examples Figure 6 are used rather than filling in the block with actual text so that the reader is given an idea of what the dataset looks like
>
> [Reply] In Figure 2, there are text representations in the background layer. Figure 6 just adds cross-mode knowledge, so the representation is highly similar to that in Figure 2. We will make better presentations of the examples.

---

> > ### Comment · Reviewer_KQtE · 2025-06-03
> >
> > Thank you for taking the time to address my concerns. I appreciate the additional data set results along with presenting the baselines to contextualize the results obtained.
> >
> > I continue to be concerned about the artificial evaluation. While measuring the models, confidence provides some signal, an LLM's confidence is not necessarily something that can be relied upon. A stronger paper would demonstrate improvement on a downstream task.

---

> > > ### Author Response · Authors · 2025-06-04
> > >
> > > Thank you for your response! We totally agree that LLMs' confidence is not trustworthy enough. The current qualitative evaluation pipeline is only a prototype. Generalization from synthetic problem to real-world tasks is definitely worth researching on. I hope our formulation and study of synthetic problem can be an important starting point for cross-mode knowledge retrieval.

---

> > > > ### Comment · Reviewer_KQtE · 2025-06-07
> > > >
> > > > I read the response and adjusted my score.

---

### Official Review · Reviewer_jyit · 2025-05-12

**Rating:** 6
**Confidence:** 3
**Ethics Flag:** 1

**Summary:**

The paper investigates cross-mode knowledge retrieval, namely whether LLMs are able to retrieve knowledge learned in one format when queried in another. The authors focus on two datasets with different textual formats - Wikipedia and Tinystories-, where they inject some random and unique token sequences, which they call knowledge pieces and are disjoint between the two datasets. In that context, knowledge retrieval is referred to as the ability to perfectly memorize such random token sequences.

Initially, they show that naively training on both datasets yields a model that has low accuracy when retrieving knowledge pieces queried in the opposite different format. Then, they explore what they refer to as dataset rewriting, where they cross-inject knowledge pieces from one mode to the other, finding that very extensive rewriting is necessary to achieve good performance. Finally, they introduce CASCADE, a training approach where they train a number of models on overlapping windows of doubling size (8, 16, … 1024 tokens) on the knowledge-injected datasets, resulting in an ensemble model. During training, loss is computed only on the second half of the context window, ensuring that every knowledge piece dominates at least half of *some* training segment. They also explore “compressing” the different trained models into one, by using a composite loss; in both cases they find significant improvements over the baseline.

**Reasons To Accept:**

- The paper identifies an interesting phenomenon: although a piece of information may be "known" by a LLM, the model struggles to accurately retrieve it when queried in a different or arbitrary format.

- I find the idea of cascading a dataset with varying context window sizes during training quite interesting. This approach introduces a curriculum-like training strategy that has the potential to disentangle knowledge from format, potentially improving performance on a variety of downstream tasks.

- The paper presents a fair amount of experimental results showing that the proposed method is able improve the cross-mode knowledge memorization performance.

- The paper is easy to follow, and every step is described in sufficient details.

**Reasons To Reject:**

- The main weakness of the paper is that the task that is being tested is purely synthetic. Although perfectly memorizing random token sequences offers a controlled setting that is reasonable for the experiments, it is unclear to me how the presented results would translate on the performance on a real downstream task. I believe that it would add great value to the author’s work if they also provided experimental evidence where CASCADE improves the performance on real downstream tasks.

- As the authors note in their conclusions, their experiments consider only two formats. While this is a reasonable starting point, I believe that exploring additional formats is necessary to fully demonstrate CASCADE’s potential. It remains unclear to me how adding more formats would affect training efficiency, and consequently, how substantial the resulting improvements would need to be to justify the added complexity.

---

> ### Author Response · Authors · 2025-06-02
> **Reply to Reviewer jyit**
>
> ### Reasons To Reject:
>
> **[R1]** The main weakness of the paper is that the task that is being tested is purely synthetic. Although perfectly memorizing random token sequences offers a controlled setting that is reasonable for the experiments, it is unclear to me how the presented results would translate on the performance on a real downstream task. I believe that it would add great value to the author’s work if they also provided experimental evidence where CASCADE improves the performance on real downstream tasks.
>
> [Reply] We agree that real-world experiments are important. As stated in Section 6, pretraining on real-world data is costly and evaluation metrics are difficult to design. While we believe the method in Appendix C reflects performance, it relies heavily on sampling from another LLM and is therefore unstable. We believe CASCADE will serve as an inspiring initial study of cross-modal knowledge retrieval.
>
>
> **[R2]** As the authors note in their conclusions, their experiments consider only two formats. While this is a reasonable starting point, I believe that exploring additional formats is necessary to fully demonstrate CASCADE’s potential. It remains unclear to me how adding more formats would affect training efficiency, and consequently, how substantial the resulting improvements would need to be to justify the added complexity.
>
> [Reply] Thanks for this valuable question -- and we conducted additional experiments on $n=3$ modes to show the efficacy of CASCADE, please read the standalone reply.

---

> > ### Comment · Reviewer_jyit · 2025-06-07
> >
> > Thank you for your response, and the additional results.
> > Although I still believe that experiments on real downstream tasks would be valuable, I agree that designing evaluation pipelines in your setting could be an independent research question on its own. Therefore, I will raise my score.

---

> > > ### Author Response · Authors · 2025-06-07
> > >
> > > Thank you for your response! We firmly believe that real-world tasks are necessary for ultimately solving the problem of cross-mode knowledge retrieval. We hope our synthetic task could serve as an initial study and insightful step towards this final goal.

---

> ### Author Response · Authors · 2025-06-05
>
> Dear Reviewer jyit,
>
> Thank you for your thoughtful feedback on our submission. We hope our response has adequately addressed your concerns. If any points require further clarification or if you have additional questions, please don't hesitate to reach out—we would be happy to provide more details.
>
> We appreciate your time and consideration.
>
> Best regards,
>
> The Authors

---

### Official Review · Reviewer_RKvZ · 2025-05-15

**Rating:** 4
**Confidence:** 3
**Ethics Flag:** 1

**Summary:**

This paper considers a problem of cross-mode knowledge retrieval where the language style used to query parametric knowledge is different from the pretraining language distribution. The authors use sentence completion as a proxy problem for demonstrating the effectiveness of their proposed CASCADE training method in the cross-model knowledge retrieval problem. While it is an interesting problem and a rather interesting approach, there are several problems:

1. The gap between sentence completion and knowledge retrieval is too big, and it is hard to imagine how success in this sentence completion problem can be transferred to the actual knowledge retrieval problem. The models that the authors trained are still causal language models, which means that the models still try to learn/memorize the association between the former and subsequent token sequences. Training with cross-mode text pairs can, of course, encourage the model to recall the sentence, but it is unclear to me how this can be generalized to knowledge that is more abstract. Ideally, based on the definition of the cross-model knowledge retrieval from the authors, I would imagine the output text sequence still carries the text style from the hint to the output while still recalling the knowledge. Perhaps, a more informative proxy problem is cross-lingual knowledge retrieval.
2. Even if I am convinced by the validity of the proxy problem, the authors did not provide enough evidence that CASCADE is more effective than using an out-of-box language model for it. There is no baseline provided in the work, which makes it hard to evaluate the effectiveness of the approach.
3. The writing of the work can be improved. The authors define a lot of notations, but only use them once or twice, which only confuses the reader. In some cases, it is actually clearer to verbally describe the method than using mathematical notations. For example, sigmoid function is well-known in the community and does not require writing out in the paper. Another example is the ratio $r=N_{OCC}/N^X_{OCC}$, which is the reciprocal of the proportion of the dataset being rewritten. Defining the ratio as the proportion (which is the reciprocal of the $r$ defined by the authors) does not change any argument but makes it more readable and intuitive.

**Reasons To Accept:**

Interesting problem; relevant to the venue.

**Reasons To Reject:**

Insufficient experiment; writing can be improved; unclear generalizability to the actual problem

---

> ### Author Response · Authors · 2025-06-02
> **Reply to Reviewer RKvZ**
>
> ### Problems:
>
> **[P1]** The gap between sentence completion and knowledge retrieval is too big, and it is hard to imagine **how success in this sentence completion problem can be transferred to the actual knowledge retrieval problem**. The models that the authors trained are still causal language models, which means that the models still try to learn/memorize the association between the former and subsequent token sequences. Training with cross-mode text pairs can, of course, encourage the model to recall the sentence, but it is unclear to me how this can be **generalized to knowledge that is more abstract**. Ideally, based on the definition of the cross-model knowledge retrieval from the authors, I would imagine the **output text sequence still carries the text style from the hint to the output while still recalling the knowledge**. Perhaps, a more informative proxy problem is **cross-lingual knowledge retrieval**.
>
> [Reply] Sentence completion is a method to **probe** knowledge, as each knowledge has a unique representation in our case, making sentence completion a suitable way to verify that the model has memorized the knowledge. For general knowledge, P-probing can possibly be applied as in [1].
>
> Maintaining the original text style is less critical than **hallucinating incorrect knowledge** -- the roadmap to fully solving cross-mode knowledge retrieval should begin by reducing the error rate. Therefore, we believe our method is a crucial step.
>
> Cross-lingual knowledge retrieval is an important problem, but we need to clarify that it is not the fundamental problem we study: If the LLM does not know the mapping between two languages, it cannot answer the question. However, if the LLM knows the mapping, it must be trained with data like (knowledge in language A, knowledge in language B), which constitutes **dataset rewriting**. In our cross-mode knowledge retrieval problem, the essential case is when two dataset modes are independent, and **no preprocessing of dataset rewriting** is done.
>
> **[P2]** Even if I am convinced by the validity of the proxy problem, the authors did not provide enough evidence that CASCADE is more effective than using an **out-of-box language model** for it. There is no baseline provided in the work, which makes it hard to evaluate the effectiveness of the approach.
>
> [Reply] For the synthetic task, since it is novel, no out-of-the-box LMs can directly solve this problem. We conducted additional experiments with a larger model, phi1-small (350M), to simulate using a more capable out-of-the-box LM. Please read the standalone reply. For the original task, based on the qualitative results in Appendix C, even GPT-4o fails, indicating that no out-of-the-box LMs can solve it either.
>
> **[P3]** The writing of the work can be improved. The authors define a lot of notations, but only use them once or twice, which only confuses the reader. In some cases, it is actually clearer to verbally describe the method than using mathematical notations. For example, sigmoid function is well-known in the community and does not require writing out in the paper. Another example is the ratio $r=N_{OCC}/N_{OCC}^X$, which is the reciprocal of the proportion of the dataset being rewritten. Defining the ratio as the proportion (which is the reciprocal of the $r$ defined by the authors) does not change any argument but makes it more readable and intuitive.
>
> [Reply] Thank you for your suggestion -- we will improve the presentation of our setting. The definition of $r=N_{occ}/N_{occ}^X$ is intended to clarify the argument that "meaningful rewriting occurs only when $r$ is close to $1$," as it covers only the range $[1, 4]$ of the spectrum $[1, 8192]$. If we set $r=N_{occ}^X/N_{occ}$, then "close to $1$" essentially covers the range $[1/4, 1]$, which represents $3/4$ of the spectrum $[0, 1]$.
>
> [1] Physics of Language Models: Part 3.1, Knowledge Storage and Extraction

---

> > ### Comment · Reviewer_RKvZ · 2025-06-03
> >
> > Thank you for the clarification.
> > I can bump up my score a bit, but at the same time, I should lower my confidence.
> > I am still not that convinced that the proxy problem is really about retrieving knowledge but just "complete this sentence".

---

> > > ### Author Response · Authors · 2025-06-04
> > >
> > > Thank you for your response!
> > >
> > > For our synthetic problem, the high level idea is we constrain everything about the knowledge into only one way, i.e., completion. So in this case, retrieving knowledge and "complete this sentence" are equivalent. This is because (1) the token range for knowledge and mode are disjoint, so knowledge and mode are not entangled, i.e., complete a random token sequence means successfully identifying "this sequence is knowledge but not mode"; (2) different knowledge have complete different random sequences, so knowledge are also mutually independent, i.e., complete a knowledge perfectly means identifying "this specific knowledge instead of (a mixture) of any other knowledge"; (3) random token sequence has no other equivalent representation compared to natural languages, i.e., there is no other way to output this knowledge. These three properties may be transribed into a formal proof using information theory.
> > >
> > > I hope this further addresses your confusion.

---

### Author Response · Authors · 2025-06-02
**Additional Experiments on More Modes and Model Sizes**

### Extension to $3$ modes

To address generalization concerns with more modes, we conducted additional experiments on $3$ modes. We used Python code from the `bigcode/the-stack` dataset as a new mode -- `code`. This dataset contains $484626190$ tokens, roughly equivalent to the other two datasets.

We present results for dataset rewriting with $r = \infty$ (direct training), $r = 1$ (full rewriting), $r = 1$ **without** $f_\text{wiki} q_\text{ts}$ in the training set, and CASCADE (with model compression) using the model (phi1-162M) specified in Appendix E.2:

| | $f_\text{ts} q_\text{ts}$ | $f_\text{wiki} q_\text{wiki}$ | $f_\text{code} q_\text{code}$ | $f_\text{ts} q_\text{wiki}$ | $f_\text{ts} q_\text{code}$ | $f_\text{wiki} q_\text{ts}$ | $f_\text{wiki} q_\text{code}$ | $f_\text{code} q_\text{ts}$ | $f_\text{code} q_\text{wiki}$ |
|-|:-:|:-:|:-:|:-:|:-:|:-:|:-:|:-:|:-:|
| Direct training ($r = \infty$) |  $-7.74 \times 10^{-6}$  | $-8.39 \times 10^{-6}$ | $-1.75 \times 10^{-5}$ | $-0.215$ | $-0.755$ | $-0.368$ | $-8.79 \times 10^{-2}$ | $-0.724$ | $-2.80 \times 10^{-2}$ |
| Full rewriting ($r = 1$) | $-2.62 \times 10^{-5}$ | $-2.78 \times 10^{-5}$ | $-3.10 \times 10^{-5}$ | $-6.89 \times 10^{-5}$ | $-1.33 \times 10^{-4}$ | $-3.46 \times 10^{-4}$ | $-8.45 \times 10^{-5}$ | $-1.51 \times 10^{-4}$ | $-1.84 \times 10^{-4}$ |
| Rewriting ($r = 1$) without $f_\text{wiki} q_\text{ts}$ | $-2.87 \times 10^{-5}$ | $-6.37 \times 10^{-6}$ | $-2.52 \times 10^{-5}$ | $-2.02 \times 10^{-4}$ | $-7.71 \times 10^{-5}$ | $-4.67 \times 10^{-3}$ (*) | $-1.37 \times 10^{-4}$ | $-1.52 \times 10^{-4}$ | $-6.32 \times 10^{-5}$ |
| **CASCADE** | $-1.97 \times 10^{-7}$ | $-1.69 \times 10^{-7}$ | $-4.24 \times 10^{-7}$ | $-1.08 \times 10^{-6}$ | $-7.01 \times 10^{-5}$ | $-8.01 \times 10^{-5}$ | $-9.42 \times 10^{-5}$ | $-6.69 \times 10^{-5}$ | $-2.82 \times 10^{-7}$ |

The key finding is that CASCADE is significantly simpler than dataset rewriting. Dataset rewriting requires rewriting all $n (n - 1) / 2 = \Theta (n^2)$ pairs of cross-mode data: in the experiment without $f_\text{wiki} q_\text{ts}$ in training, evaluation performance on $f_\text{wiki} q_\text{ts}$ is worse than in all other cases. CASCADE's simple algorithmic nature allows it to generalize easily to more modes **without any extra effort**, making it advantageous over the baselines.

### Increasing model size

We also conducted experiments using phi1-small (350M) [1], with results reported below:

| | $f_\text{ts} q_\text{ts}$ | $f_\text{wiki} q_\text{wiki}$ | $f_\text{code} q_\text{code}$ | $f_\text{ts} q_\text{wiki}$ | $f_\text{ts} q_\text{code}$ | $f_\text{wiki} q_\text{ts}$ | $f_\text{wiki} q_\text{code}$ | $f_\text{code} q_\text{ts}$ | $f_\text{code} q_\text{wiki}$ |
|-|:-:|:-:|:-:|:-:|:-:|:-:|:-:|:-:|:-:|
| Direct training ($r = \infty$) |  $-6.73 \times 10^{-7}$  | $-9.03 \times 10^{-7}$ | $-6.25 \times 10^{-7}$ | $-4.19 \times 10^{-2}$ | $-0.277$ | $-5.54 \times 10^{-2}$ | $-1.39 \times 10^{-2}$ | $-0.172$ | $-3.80 \times 10^{-3}$ |
| Full rewriting ($r = 1$) | $-5.68 \times 10^{-6}$ | $-5.80 \times 10^{-6}$ | $-6.25 \times 10^{-6}$ | $-2.21 \times 10^{-5}$ | $-2.89 \times 10^{-5}$ | $-6.16 \times 10^{-5}$ | $-5.51 \times 10^{-5}$ | $-3.12 \times 10^{-5}$ | $-1.65 \times 10^{-5}$ |

The key finding is that increasing model size from 162M to 350M improves each method's performance by approximately one order of magnitude. With this increase, full rewriting (350M) can match CASCADE (162M), while direct training still struggles with cross-mode knowledge retrieval.

[1] Textbooks Are All You Need

---

### Decision · Program_Chairs · 2025-07-08

**Decision:**

Accept

**Comment:**

**Pros**

Clarity:
- The paper is well-written and easy to follow.

Originality and Significance:
- The authors identify an interesting and unique problem. This may be the first paper to recognize this problem and propose an efficient method to address it.

**Cons**

Clarity:
- The appendix contains too many points. The authors should reorganize the paper to integrate the additional content into the main sections, utilizing the extra pages available in the updated version.
- It is worth noting that the knowledge experiment conducted is somewhat limited and does not fully reflect typical information retrieval tasks for knowledge retrieval.

Significance:
- Including real-data experiments would strengthen the paper.
- Adding the author response experiments with n=3 and a larger model would further support the paper's claims.

Overall, reviewers agree that this paper addresses an interesting and novel problem that is gaining increasing attention. The proposed approach is well-suited for the problem, as evidenced by the results in the paper. For the updated version, it would be helpful to reorganize the appendix content into the main sections and include the additional experiments with n=3 as discussed in the author response.